# IKK2/NFkB signaling controls lung resident CD8+ T cell memory during influenza infection

Curtis J. Pritzl [1,2], Dezzarae Luera [1,2], Karin M. Knudson[1], Michael J. Quaney[1], Michael J. Calcutt[3], Mark A. Daniels [1,2] & Emma Teixeiro [1,2] ✉

CD8+ T cell tissue resident memory ($T_{RM}$) cells are especially suited to control pathogen spread at mucosal sites. However, their maintenance in lung is short-lived. TCR-dependent NFkB signaling is crucial for T cell memory but how and when NFkB signaling modulates tissue resident and circulating T cell memory during the immune response is unknown. Here, we find that enhancing NFkB signaling in T cells once memory to influenza is established, increases pro-survival Bcl-2 and CD122 levels thus boosting lung CD8+ $T_{RM}$ maintenance. By contrast, enhancing NFkB signals during the contraction phase of the response leads to a defect in CD8+ $T_{RM}$ differentiation without impairing recirculating memory subsets. Specifically, inducible activation of NFkB via constitutive active IKK2 or TNF interferes with TGFβ signaling, resulting in defects of lung CD8+ $T_{RM}$ imprinting molecules CD69, CD103, Runx3 and Eomes. Conversely, inhibiting NFkB signals not only recovers but improves the transcriptional signature and generation of lung CD8+ $T_{RM}$. Thus, NFkB signaling is a critical regulator of tissue resident memory, whose levels can be tuned at specific times during infection to boost lung CD8+ $T_{RM}$.

Once infection has resolved, a few of the pathogen-specific T cells that participated in the response persist as memory cells providing the host with enhanced protection against re-infection[1–3]. These memory T cells ($T_{MEM}$) strategically relocate to blood and secondary lymphoid organs (central, $T_{CM}$ and effector, $T_{EM}$ memory) as well as portal of entry tissues (tissue resident, $T_{RM}$) each, with specific phenotypes and functions[4]. Together, they guarantee the generation of a diverse and polyfunctional memory pool. In contrast to other memory subsets, $T_{RM}$ cells do not leave the tissue, and continue patrolling it for signs of pathogen re-entry. If this happens, they trigger innate immune responses and immediately control reinfection in situ, in tissues like lung, skin or gut[5]. $T_{RM}$ cells have a protective role not only in infectious diseases[6–9], but also in cancer[10–13]. Yet, mounting evidence also associates $T_{RM}$ with pathology in autoimmunity, transplants, and graft versus host disease[14–16]. Although this puts $T_{RM}$ as a therapeutic target

to treat disease, there is still a lack of understanding of how $T_{RM}$ cells are generated and maintained in tissues. Furthermore, the times during the immune response that are suitable for manipulation of $T_{RM}$ for therapeutic purposes are ill defined. This is particularly important in the case of respiratory infections such as influenza that depend on lung CD8+ $T_{RM}$ to control viral titers and disease severity[17,18] but where CD8+ $T_{RM}$ longevity is limited[18].

One of the cardinal features of $T_{RM}$ cells is their imprinting of non-lymphoid "tissue residency", which differentiates them from circulating memory T cells ($T_{CIRCM}$). This is phenotypically characterized by high expression of CD69 and often (but not always) CD103. Transcriptionally, CD8+ $T_{RM}$ cells require high expression of Runx3[11], Nur77[19,20] and low expression of Eomes[21], although depending on the tissue, a balanced expression of other transcription factors, such as Blimp1 in lung[22], is also important. Signals that

[1]Department of Molecular Microbiology and Immunology, School of Medicine, University of Missouri, Columbia, MO, USA. [2]Roy Blunt NextGen Precision Health Building, School of Medicine, University of Missouri, Columbia, MO, USA. [3]Department of Veterinary Pathobiology, University of Missouri, Columbia, MO, USA. ✉e-mail: teixeiropernase@missouri.edu

occur prior to tissue entry[23] and tissue-specific signals[24] both contribute to the differentiation of $T_{RM}$. Among these, antigen and TGFβ signals act at different points of the immune response to shape $T_{RM}$[25–31]. Yet, the role of inflammation in the generation and maintenance of $T_{RM}$ remains largely unexplored.

NFkB signaling is a major driver of inflammation[32,33] as well as one of the signaling pathways induced by T cell receptor signaling upon antigen recognition[34]. Multiple pro-inflammatory factors (such as TNF, IL-1, or TLRs), together with antigen, signal through the canonical NFkB pathway at different times during infection[35–37], making it a plausible signaling hub where different environmental cues converge to regulate T cell differentiation and cell fate decisions. Here we sought to understand how changes in the levels of IKK2/NFkB signaling that a CD8[+] T cell experiences during infection, impact their memory fate. Our data show that NFkB signaling has a specific role in tissue-resident memory that is different from the other recirculating memory subsets. Furthermore, NFkB signaling differentially regulates CD8[+] $T_{RM}$ differentiation and CD8[+] $T_{RM}$ maintenance. Interestingly, our data also reveal that tuning NFkB signaling levels at specific times during influenza infection can aid to boost or deplete CD8[+] $T_{RM}$ in the lung, an organ where these cells gradually vanished over time after vaccination or infection, leading to loss in protection[38].

## Results

### Increasing the levels of NFkB signaling after the peak of the immune response to influenza infection improves circulating CD8[+] T cell memory

To address the impact of NFkB signaling on T cell protective immunity, we generated two tetON inducible systems restricted to the T cell lineage. For this, we crossed mice carrying either a constitutively active Ikbkb allele (CA-IKK2)[39] or a dominant negative-acting version of IKK2 (DN-IKK2)[40] driven by the tetracycline TA-activated promoter (tetO)7 transactivator with mice expressing CD2-driven rtTA[41]. We refer to these mice as CD2rtTAxCA-IKK2 and CD2rtTAxDN-IKK2, respectively (Supplementary Fig. 1 and 2). Expression of CA- and DN-IKK2 can be monitored by a luciferase reporter (either by flow cytometry or by luciferase assays) and is restricted to the T cell lineage (Supplementary Fig. 2b, c). Furthermore, doxycycline (DOX) dependent induction of IKK2 (CA-IKK2[ON]) results in the upregulation of NFkB-dependent genes[42,43] Supplementary Fig. 2d). Importantly, in these inducible models, constitutive activation of IKK2 in vivo does not lead to overt T cell apoptosis (no induction of cleaved caspase-3 or FasL) (Supplementary Fig. 2d).

Next, we used these two inducible models to interrogate whether changing the levels of IKK2/NFkB signaling in T cells during specific phases of the immune response impacts CD8[+] T cell memory. We tested whether boosting (or inhibiting) IKK2/NFkB signal transduction could modulate the establishment of circulating CD8[+] memory in two different polyclonal models of infection. For this, we used both tetON IKK2 inducible models and manipulated NFkB signaling following on previous reports that suggested a role for p65NFkB transcriptional activity during the contraction phase of the immune response to *Listeria monocytogenes*[42,44]. We found that inhibition of NFkB signaling (DN-IKK2[ON]), during the contraction phase of the response, led to a loss of circulating CD8[+] T cell memory. By contrast, increasing NFkB signaling (CA-IKK2[ON]) resulted in a considerable increase in the number of polyclonal antigen-specific memory CD8[+] T cells generated, both against influenza A virus (IAV) and vesicular stomatitis virus (VSV) infection (Supplementary Fig. 2e–h). Thus, NFkB has a critical role in the establishment of CD8[+] T cell memory upon infection and most importantly, changing the levels of NFkB signaling can be used to boost or reduce CD8[+] T cell memory.

### IKK2/NFkB signaling differentially regulates T cell memory subset diversity

Since CD8[+] T cell memory is composed of different subsets with unique locations and phenotypes, we next asked whether the impact of IKK2/NFkB signaling would equally affect all T cell memory subsets. Using our inducible models, we infected mice with IAV X31 and allowed CD8[+] T cells to differentiate for 5 days. After this time, we either increased or decreased NFkB signaling in T cells by feeding mice with doxycycline chow for 25 additional days and then, measured the frequency and number of IAV-specific CD44[hi]CD62L[hi] (central memory or $T_{CM}$), CD44[hi] CD62L[lo] (effector memory or $T_{EM}$) and $T_{RM}$ generated (Fig. 1a). CD44[hi] CD62L[hi] and CD44[hi] CD62[lo] $T_{MEM}$ were measured in draining lymph nodes while $T_{RM}$ was assessed by intravascular (IV) staining, a method widely accepted in the field to identify T cells in the lung parenchyma[5]. Inhibition of NFkB signaling during the contraction phase of the IAV immune response, impaired the generation of both CD44[hi] CD62L[hi] and CD44[hi] CD62L[lo] influenza-specific CD8[+] $T_{MEM}$ cells. However, increasing NFkB signaling, improved the numbers of IAV-specific $T_{CM}$ generated but not CD44[hi]CD62L[lo] $T_{MEM}$ cells (Fig. 1b, c). CD44[hi] CD62L[lo] $T_{MEM}$ cells are often identified as effector memory ($T_{EM}$) cells[45]. However, $T_{EM}$ cells are more prominent in peripheral tissues than in lymph nodes. Therefore, it is possible that the CD44[hi] CD62[lo] CD8[+] $T_{MEM}$ population we examined included a substantial number of lymphoid CD69[+] CD103[+] $T_{RM}$ phenotype cells[46,47]. Interestingly, when we examined the generation of lung $T_{RM}$, we observed that NFkB signaling had opposite effects on this T cell subset than on the other $T_{MEM}$ subsets present in the mediastinal lymph nodes. Generation of polyclonal IAV NP[366] and PA[224] antigen-specific CD8[+] $T_{MEM}$ resident in the lung parenchyma (IV negative) was decreased under high levels of NFkB signals, both in frequencies and numbers. On the contrary, decreasing NFkB signaling resulted in a dramatic boost in the number of lung polyclonal IAV (NP and PA) specific CD8[+] $T_{RM}$ generated (Fig. 1d, e). These differences were maintained when comparing the frequency and number of influenza-specific cells in the lung parenchyma that co-expressed the canonical $T_{RM}$ markers CD69 and CD103 (Fig. 1f). Importantly, our observations were not exclusive to the lung or the type of infection as the same results were observed for antigen-specific CD8[+] $T_{RM}$ generated in the kidney upon VSV infection (Fig. 1g and Supplementary Fig. 3).

It is also key to note that increasing NFkB levels did not result in an overall decrease in the number of total or parenchyma resident CD8[+] or CD4[+] T cells in the lung, indicating that the effects of NFkB were on antigen specific CD8[+] T cells (Fig. 2b, c). We also did not observe differences in the frequency of lung IAV-specific CD4[+] memory T cells when NFkB signals were increased, suggesting that the defect in CD8[+] $T_{RM}$ was not due to decreased or increased generation of influenza-specific CD4[+] T cells (Supplementary Fig. 4d).

We also investigated whether impaired generation of lung CD8[+] $T_{RM}$ caused by enhanced NFkB signaling could be due to overt inflammation in the lung. A blinded independent pathologist evaluated inflammation in lung sections of mice where constitutive NFkB signaling was induced from day 5 to day 30 upon DOX administration and compared it to controls (no DOX). No signs of parenchymal disruption were apparent neither in control nor in mice subjected to DOX (Supplementary Fig. 4a). Increased NFkB signaling did not lead to non-specific activation of NK cells or dendritic cells (DC) either (Supplementary Fig. 4b, c).

Finally, we used an adoptive transfer model to assess whether the effect of NFkB signaling was CD8[+]T cell intrinsic. In response to both, influenza and VSV intranasal (i.n.) infection, we found that increasing NFkB signaling late in the response (contraction phase), and only in CD8[+] T cells, led to a severe loss in antigen-specific polyclonal and monoclonal lung CD8[+] $T_{RM}$ (Fig. 2d, e). In contrast to this, inhibiting NFkB signaling intrinsically in CD8[+] T cells, led to an increase in the

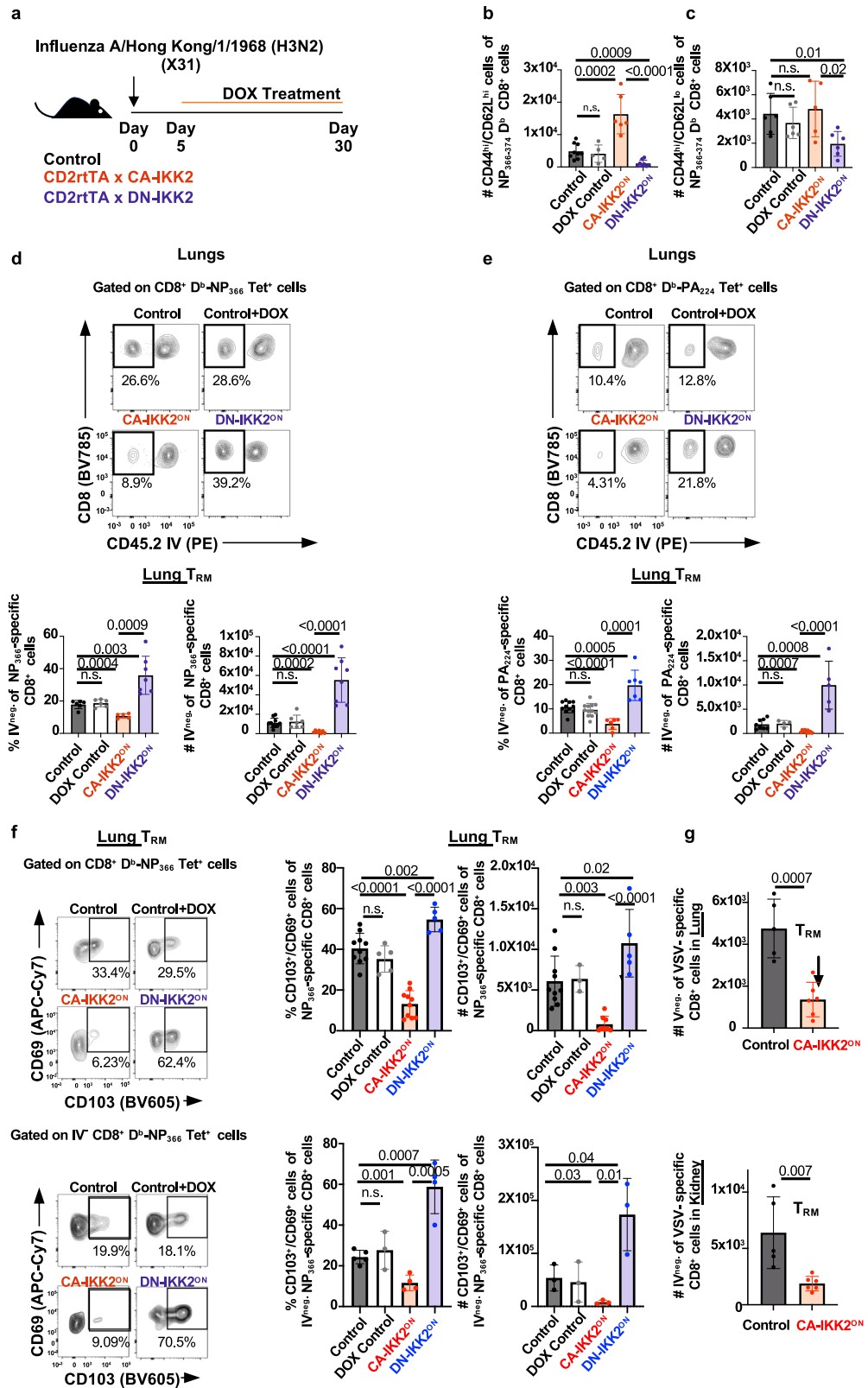

generation of influenza-specific lung CD8$^+$ T$_{RM}$ (Fig. 2f). Altogether data from Figs. 1, 2 reveal that IKK2/NFkB signaling is a critical pathway in the regulation of CD8$^+$ T cell memory subset diversity. Furthermore, inhibition of NFkB signals can boost lung CD8$^+$ T$_{RM}$, whereas CD8$^+$ T cells that are intrinsically exposed to high levels of NFkB signaling are impaired in the generation of protective memory T cells resident in the lung.

## Increasing IKK2 signaling impairs protection against heterologous infections

CD8$^+$ T$_{RM}$ is critical to provide protection in tissues against re-infection[18,48]. We, thus, tested whether increasing the amount of NFkB signaling in CD8$^+$ T cells as they differentiate to T$_{RM}$ would impact protective immunity in the lung. For this, we followed a published approach to deplete circulating CD8$^+$ T cells while sparing CD8$^+$ T$_{RM}$[49].

**Fig. 1 | NFkB signaling differentially regulates T cell memory subset diversity. a** Control, CD2rtTA x CA-IKK2 or CD2rtTA x DN-IKK2 mice were infected with influenza X31(1000 pfu). From 5–30 days post-infection (d.p.i.), mice were fed a DOX or a control diet. Numbers (#) in mediastinal lymph nodes at 30 d.p.i of (**b**) NP$_{366-374}$-specific CD44$^{hi}$, CD62L$^{hi}$ T$_{MEM}$ cells (CD8$^+$, Db-NP-tet$^+$, CD44$^+$, CD62L$^+$), $n = 9$ (control), $n = 5$ (DOX control), $n = 6$ (CA-IKK2$^{ON}$), $n = 8$ (DN-IKK2$^{ON}$) mice pooled from 3 independent experiments (**c**) NP$_{366-374}$-specific CD44$^{hi}$, CD62L$^{lo}$ (CD8$^+$ Db-NP-tet$^+$ CD44$^{hi}$CD62L$^{lo}$) T$_{MEM}$ cells. $n = 6$ (control), $n = 6$ (DOX control), $n = 5$ (CA-IKK2$^{ON}$), $n = 6$ (DN-IKK2$^{ON}$) mice pooled from 4 independent experiments. (**d–g**) Lung T$_{RM}$ were identified in X31-infected mice by intravascular (IV) staining with CD45.2 to discriminate vascular (IV. CD45.2$^+$) and parenchyma resident (IV CD45.2$^-$). The frequency (%) and numbers of influenza (**d**) NP$_{366-374}$ specific CD8$^+$ T$_{RM}$. For %, #: $n = 6, 10$ (control), $n = 6, 7$ (DOX control), $n = 5, 7$ (CA-IKK2$^{ON}$), $n = 7, 8$ (DN-IKK2$^{ON}$) mice pooled from 2 independent experiments. **e** PA$_{224-233}$ specific

CD8$^+$ T$_{RM}$. $n = 11, 8$ (control), $n = 12, 4$ (DOX control), $n = 6, 9$ (CA-IKK2$^{ON}$), $n = 7, 5$ (DN-IKK2$^{ON}$) mice pooled from 2 independent experiments. **f** Representative dot plots showing frequencies and numbers of cells with the CD69$^+$, CD103$^+$ phenotype among NP$_{366-374}$ specific CD8$^+$ T cells (top graphs) or among IV negative (bottom graphs). For %, #: $n = 10, 10$ (control), $n = 5, 3$ (dox control), $n = 10, 8$ (CA-IKK2$^{ON}$), $n = 5, 5$ (DN-IKK2$^{ON}$) mice pooled from $n = 2$ independent experiments for top graphs. $n = 5, 3$ (control), $n = 3, 3$ (DOX control), $n = 4, 3$ (CA-IKK2$^{ON}$), $n = 4, 3$ (DN-IKK2$^{ON}$) mice pooled from 2 independent experiments for bottom graphs. **g** Control or CD2rtTA x CA-IKK2 (CA-IKK2$^{ON}$) mice were infected with $2 \times 10^6$ pfu VSV, fed a DOX-containing diet from 5 to 30 d.p.i. At 30 d.p.i. VSV-specific CD8$^+$ T$_{RM}$ (Kb-N-tet$^+$ CD8$^+$ CD45.2$^-$) were identified in the lungs and kidneys of infected mice by IV staining. $n = 5$ (control), $n = 6$ (CA-IKK2$^{ON}$) mice pooled from 2 independent experiments. Bars represent mean values ± SD. *P*-values were determined by a two-tailed unpaired *t* test, n.s. not significant. Source data provided as a source data file.

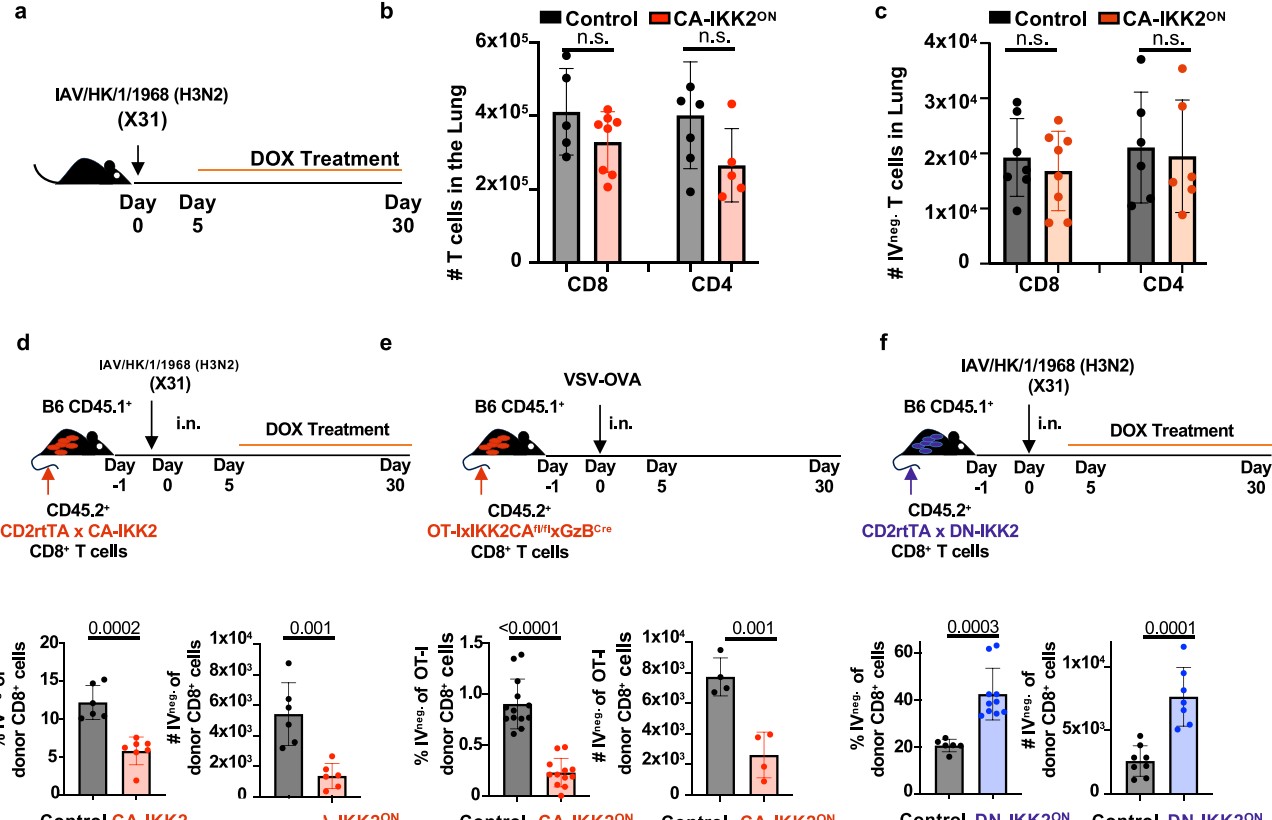

**Fig. 2 | Impairment in the generation of CD8$^+$ T$_{RM}$ under enhanced NFkB signals is T cell intrinsic. a–c** Control or CD2rtTA x CA-IKK2 mice were infected intranasally (i.n.) with influenza X31 (15000 pfu). From 5 to 30 d.p.i., mice were fed a DOX (CA-IKK2$^{ON}$) or control diet. Graph shows number of total (**b**) and parenchymal (**c**) CD8$^+$ and CD4$^+$ T cells in the lungs. (**b**) for CD4$^+$ T cells: $n = 5$ (control), $n = 8$ (CA-IKK2$^{ON}$); for CD8$^+$ T cells $n = 6$ (control), $n = 5$ (CA-IKK2$^{ON}$); (**c**) for CD4$^+$ T cells: $n = 6$ (control and CA-IKK2$^{ON}$); for CD8$^+$ T cells $n = 7$ (control), $n = 8$ (CA-IKK2$^{ON}$) mice pooled from 2 independent experiments. **d** Naïve donor CD8$^+$ T cells from CD2rtTAxCA-IKK2 mice were adoptively transferred to congenic hosts which were next infected with influenza X31(1000 pfu). Host mice were fed DOX or control diets from 5 to 30 d.p.i. At 30 d.p.i., frequencies and numbers of lung CD8$^+$ donor T$_{RM}$ cells were determined by IV neg. labeling. For %: $n = 6$ (control), $n = 7$ (CA-IKK2$^{ON}$); For #: $n = 6$ (control and CA-IKK2$^{ON}$) mice pooled from 2 independent

experiments. **e** Naïve CD8$^+$ OT-I T cells isolated from OT-IxIKK2CA$^{fl/fl}$xGzB$^{Cre}$ mice were adoptively transferred to congenic host mice followed by i.n. VSV-OVA infection. At 30 d.p.i., lung resident OT-I donor frequencies and numbers were determined by IV neg. staining. For %: $n = 13$ (control), $n = 12$ (CA-IKK2$^{ON}$); mice pooled from 3 independent experiments. For num#: $n = 4$ (control and CA-IKK2$^{ON}$) mice pooled from 2 independent experiments. **f** CD2rtTA x DN-IKK2 naïve CD8$^+$ T cells were transferred to congenic host mice. Hosts were infected with influenza X31. Beginning 5 d.p.i., either a DOX or a control diet was administered. At 30 d.p.i. donor CD8$^+$ lung T$_{RM}$ cell frequencies and numbers were determined by IV neg. staining. $n = 6$ (control), $n = 10$ (DN-IKK2$^{ON}$) mice pooled from 2 independent experiments. Bars represent mean values ± SD. *P*-values were determined by a two tailed unpaired *t* test, n.s. not significant. Source data provided as a source data file.

We adoptively transferred OT-I naïve male T cells into female or male congenically marked hosts, followed by intranasal VSV-OVA infection (Fig. 3a). Consistent with rejection against male antigen, male circulating donors vanished in female but not in male hosts (Fig. 3b, c and Supplementary Fig. 5a). These conditions allowed us to singularly

evaluate the T cell protective ability of antigen specific lung CD8$^+$ T$_{RM}$ generated under high levels of NFkB signaling in the context of heterologous intranasal PR8-OVA infection (Fig. 3a). Consistent with our results in Fig. 2, high levels of NFkB signaling (CA-IKK2$^{ON}$ model) led to a loss of CD8$^+$ T$_{RM}$ (Fig. 3d, Supplementary Fig. 5b). Most importantly,

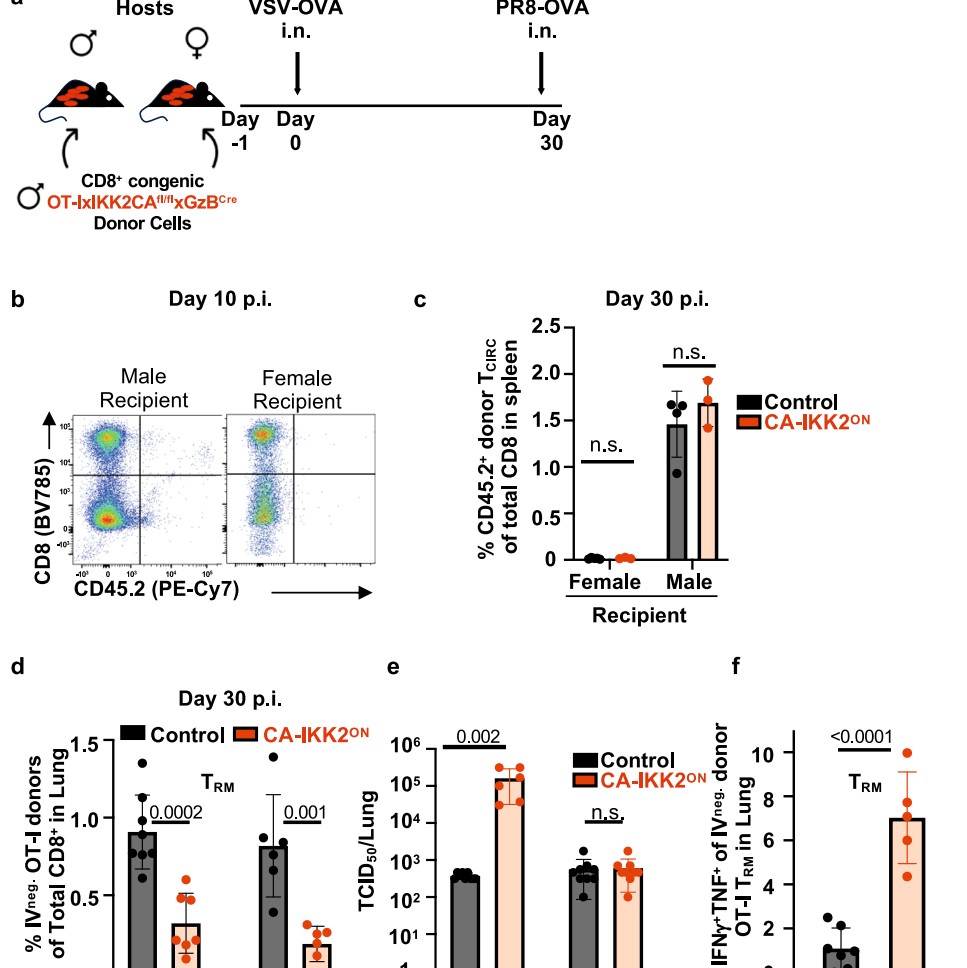

**Fig. 3 | Enhancing NFkB signaling reduces $T_{RM}$ and compromises protection against heterologous infection. a** Naive CD8+ donor T cells from male OT-IxIKK2CA$^{fl/fl}$xGzB$^{Cre}$ or OT-I littermate control mice were adoptively transferred into groups of congenically marked male and female host mice, followed by intranasal infection with $2 \times 10^4$ pfu VSV-OVA. **b** Rejection was determined at 10 p.i. in mediastinal lymph nodes. Representative dot plots are shown. **c** Graph shows frequency of donor CD8+ T cells in the spleen at 30 d.p.i. Female ($n = 5$, control; $n = 3$, CA-IKK2$^{ON}$) and male ($n = 4$, control; $n = 3$, CA-IKK2$^{ON}$) mice from a representative of 2 independent experiments. **d** Graph shows the frequency of lung parenchyma resident, donor OT-I CD8+ T cells determined by intravascular staining in female and male hosts at 30 d.p.i. Graph shows combined data from female ($n = 8$, control;

$n = 7$, CA-IKK2$^{ON}$) and male ($n = 6$, control and CA-IKK2$^{ON}$) mice pooled from 2 independent experiments. **e, f** A subset of mice from each group was challenged with influenza PR8-OVA (5000 pfu). **e** Virus titers were determined 2.5 days post-challenge in homogenized lungs. Combined data from female ($n = 9$, control; $n = 6$, CA-IKK2$^{ON}$) and male ($n = 9$, control and CA-IKK2$^{ON}$) mice pooled from 2 independent experiments (**f**) Frequency of OT−1 CD8+ donor $T_{RM}$ expressing both IFNγ and TNF at 30 d.p.i. upon ex vivo OVA antigen stimulation. Combined data from $n = 7$, control; $n = 5$, CA-IKK2$^{ON}$ mice from 2 independent experiments. Bars represent mean values ± SD. P-values were determined by a multiple unpaired t test (**c, e**), two-tailed unpaired t test (**d, f**), n.s. not significant Days post-infection or d.p.i. Source data provided as a source data file.

female hosts bearing only male CA-IKK2$^{ON}$ CD8+ $T_{RM}$ cells exhibited ~200 times higher virus titers than their control counterparts in the lung upon influenza infection (Fig. 3e). This was despite better effector T cell memory function (Fig. 3f). These data thus show that loss of CD8+ $T_{RM}$ due to high levels of NFkB signaling impairs protective immunity in the lung.

**IKK2/NFkB signaling interferes with late CD8+ $T_{RM}$ transcriptional programming**

Recent reports suggest that CD8+ $T_{RM}$ development results from a combination of signals that T cells receive before tissue entry and signals that occur later, in tissue[23,50–52]. In our model, NFkB signals were over-induced after several days of infection coinciding with a time where some antigen-specific effector T cells are being recruited to tissue while others are continuing their differentiation in tissue. We

thus sought to determine when CD8+ $T_{RM}$ loss began for CA-IKK2$^{ON}$ CD8+ T cells. For this, we repeated similar experiments to the ones in Fig. 1 and monitored the frequency and number of influenza-specific CA-IKK2$^{ON}$ and control CD8+ CD69+ T cells in lung at day 10 p.i and 30 p.i. (Fig. 4a–c, h). We found that the loss of CA-IKK2$^{ON}$ CD8+ $T_{RM}$ happened between day 10 and day 30 p.i. (Fig. 4b, c and Supplementary Fig. 6). The loss of CD8+ $T_{RM}$ correlated with a reduction in the number of CD8+ $T_{RM}$ expressing the $T_{RM}$ tissue markers CD69 and CD103 (Fig. 4c, d and Supplementary Fig. 6). However, we did not observe any defect in the frequency or number of circulating CD8+ memory T cells in the lung (Fig. 4f) or in the expression of CXCR3, one of the chemokine receptors important for lung recruitment (Fig. 4e)[53]. Interestingly, when we assessed the frequency and number of CD69 negatives within the IV negative CD8+ population, we observed that CA-IKK2$^{ON}$ CD8+ T cells lost CD69 expression but appeared in the parenchyma in

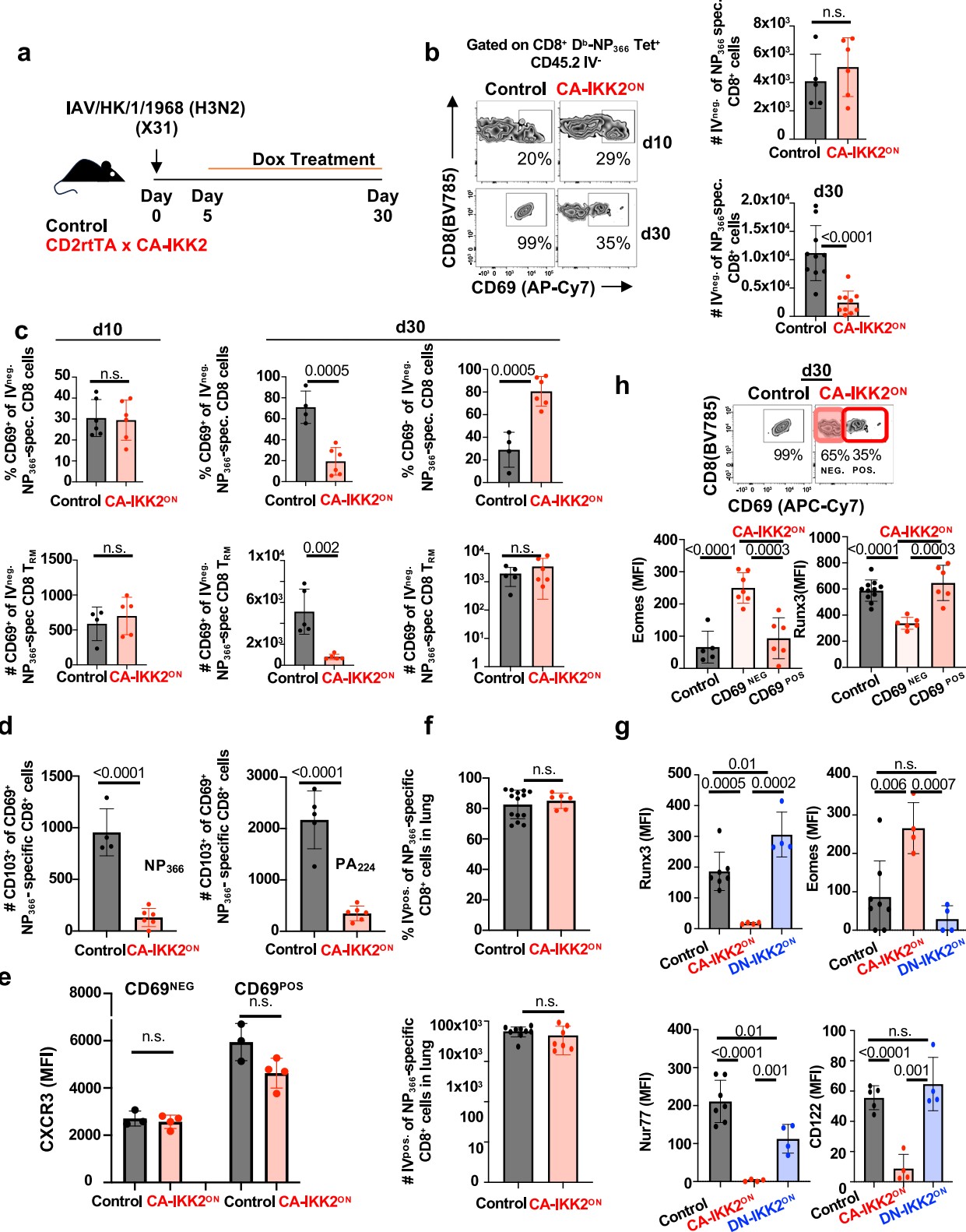

numbers close to controls (as opposed to disappearing due to cell death or quickly abandoning the parenchyma), suggesting an increase in the frequency of trafficking "effector" memory CD8$^+$ population (Fig. 4c, day 30, graphs on the right). Therefore, enhanced NFkB signals do not interfere with the recruitment of CA-IKK2$^{ON}$ cells to the lung but rather with the ability of these cells to stay in parenchyma as "bona fide" T$_{RM}$ (CD69$^+$).

To further investigate this possibility, we tested whether the defect in the establishment of the CA-IKK2$^{ON}$ CD8$^+$ T$_{RM}$ pool was a consequence of T$_{RM}$ transcriptional programming and/or survival. CD8$^+$ T$_{RM}$ differentiation requires downregulation of T-box transcription factors Eomes and T-bet[21,54], induction of Runx3 and Nur77[11,55] and Blimp1 in the lung[22]. Additionally, some studies attribute a role for IL-15 in the homeostasis/survival of CD8$^+$ T$_{RM}$ in

**Fig. 4 | NFkB signaling regulates CD8+ T$_{RM}$ transcriptional programming.**
**a, b** Control or CA-IKK2$^{ON}$ mice were infected with X31(1000 pfu) and ±DOX day 5–30 p.i. Influenza-specific CD8+ T cells (Db-NP$_{366–374}$ or Db-PA$_{224–233}$ tetramers+) in the lung parenchyma (T$_{RM}$). **b** Representative plots of CD69+ cells among lung NP$_{366–374}$+ CD8+ T cells. Frequencies (%) and numbers (#) of NP$_{366–374}$+ CD8+ T cells at day 10 (top graph) and day 30 (bottom) p.i. Day 10: $n = 5$, control; $n = 6$ CA-IKK2$^{ON}$; Day 30: $n = 10$, control and CA-IKK2$^{ON}$ mice pooled from 3 independent experiments. **c** % and # of CD69+ (left and middle) and CD69− (right) cells among the lung NP$_{366–374}$ specific CD8+ T$_{RM}$ at day 10 (left) and day 30 (right) p.i. Day 10, for %: $n = 6$ (Control and CA-IKK2ON); for #: $n = 4$ (control), $n = 5$ (CA-IKK2$^{ON}$). For Day 30 CD69$^{+/-}$ cells: for % of $n = 4$, (control), $n = 6$ (CA-IKK2$^{ON}$); for # $n = 5$ mice from 2 independent experiments. **d** Numbers of CD103+ cells among the CD69+ cells. NP specific (left) and PA specific (right) at day 30 p.i. For NP: $n = 4$, control; $n = 6$ CA-IKK2$^{ON}$; For PA$_{224}$: $n = 5$, control; $n = 6$ CA-IKK2$^{ON}$ mice from 2 independent

experiments. **e** CXCR3 expression in CD69− and CD69+ populations among the NP-specific T$_{RM}$ at day 30 p.i. $n = 3$, control; $n = 4$, CA-IKK2$^{ON}$ mice from 2 independent experiments. **f** % and # of circulating NP specific CD8+ T cells at day 30 p.i. For %: $n = 14$, control, $n = 6$ CA-IKK2$^{ON}$; For #: $n = 9$, control, $n = 7$ CA-IKK2$^{ON}$ mice from 3 independent experiments. **g** Expression of the indicated T$_{RM}$-associated transcription factors and CD122 in influenza-specific CD8+ T$_{RM}$ at 30 d.p.i. Control: $n = 7$ (Runx3 and Nur77), $n = 8$ (Eomes), $n = 5$(CD122); for CA-IKK2$^{ON}$ and DN-IKK2$^{ON}$ $n = 4$ mice from 3 independent experiments. **h** Eomes and Runx3 expression was determined in CD69$^{+/-}$ cells among the NP-specific CD8+ T$_{RM}$ at 30 d.p.i. as indicated. CD69: $n = 5, 7, 6$ (control; CD69+, CD69−). Eomes: $n = 11$, control; CD69$^{+/-}$: $n = 6$ mice pooled from 2 independent experiments. Bars represent mean values ± SD. P-values were determined by two-tailed unpaired t test, n.s. not significant. Source data provided as a source data file.

tissue[26,56,57]. We observed that CA-IKK2$^{ON}$ CD8+ T$_{RM}$ expressed higher levels of T-bet and Eomes than their control counterparts. However, they exhibited reduced levels of Nur77 and Runx3 and expressed normal levels of Blimp1 (Fig. 4g and Supplementary Fig. 6e, f). Conversely, DN-IKK2$^{ON}$ CD8+ T$_{RM}$ exhibited a reversion of the levels of Nur77 and Eomes and an increase in the induction of Runx3 over control levels (Fig. 4g). The expression of CD122, one of the chains of the IL-15R, was also impaired in CA-IKK2$^{ON}$ CD8+ T$_{RM}$ cells but not in DN-IKK2$^{ON}$ CD8+ T$_{RM}$ cells. (Fig. 4g). Importantly, CD69 negative CA-IKK2$^{ON}$ CD8+T cells in the parenchyma also exhibited defects in the expression of T$_{RM}$ transcription factors Eomes and Runx3 when compared with their CD69 positive counterparts and controls (Fig. 4h). Collectively, these data support the idea that the level of NFkB signaling that a CD8+ T cell experiences can shape its T$_{RM}$ transcriptional programming and its survival within the lung parenchyma.

## NFkB signaling inhibits TGFβ signals for CD8+ T$_{RM}$

The fact that the number of CA-IKK2$^{ON}$ CD8+ T$_{RM}$ cells started to decrease late in the immune response and that this coincided with changes in T$_{RM}$-associated transcription factors (Eomes and Runx3) that are regulated by tissue cues[24,50], led us to hypothesize that NFkB signaling could be inhibiting tissue signals that are required for lung CD8+ T$_{RM}$ differentiation. TGFβ is a crucial cytokine for tissue-resident memory differentiation and plays a specific role in lung. Moreover, both Runx3 and CD103 are downstream targets of TGFβ signaling[58,59], and our data showed defects in both T$_{RM}$ markers in CA-IKK2$^{ON}$ CD8+ T$_{RM}$ cells (Figs. 1f, 4g, h). Therefore, we tested whether NFkB signaling could inhibit TGFβ signaling in differentiating CD8+ T cells. For this, we used our two inducible models where IKK2 signaling can be increased in CD8+ T cells (Fig. 2d, e). CD8+ T cells that were exposed to TGFβ while NFkB signaling was over-induced indeed expressed lower levels of CD103 and Runx3 than TGFβ controls (Fig. 5a). Furthermore, increasing NFkB signaling also resulted in defective canonical (phosphorylated Smad2/3 or p-Smad2/3) and non-canonical (phosphorylated ERK or p-ERK) TGFβ signal transduction (Fig. 5b, d, Supplementary Fig. 7). Of note, in the context of infection, CA-IKK2$^{ON}$ CD8+ T cells experiencing high levels of NFkB signaling also exhibited lower levels of TGFβ signaling than their control counterparts (Fig. 5c).

In other cell types, NFkB can interfere with TGFβ signals through the expression of the inhibitory protein Smad7[60]. Thus, we next determined the levels of Smad7 under the conditions in Fig. 5a. Consistent with the idea that NFkB signals could induce Smad7 expression in T cells and thereby inhibit TGFβ signaling, we observed that CD8+ T cells under high NFkB signaling upregulated Smad7 expression as p-Smad2/3 and p-ERK levels decreased (Fig. 5d, Supplementary Fig. 7). Collectively, these data show that IKK2/NFkB signaling negatively regulates TGFβ signaling and the CD8+ T$_{RM}$ markers, Runx3 and CD103.

## TNF-mediated NFkB signaling inhibits TGFβ signaling for T$_{RM}$

NFkB signaling is a mediator of inflammatory signals elicited during respiratory infections[61–63]. One of the most understood NFkB triggers is the pro-inflammatory cytokine TNF, which has been associated with T$_{RM}$ in the lung[64] and signals through the canonical NFkB pathway[35]. Therefore, we hypothesized that TNF could, via NFkB, inhibit TGFβ signaling and impair CD8+ T$_{RM}$. To directly test this, we designed an experiment where CD8+ T cells differentiating in the presence of TNF were exposed to TGFβ and then measured changes in TGFβ signaling as well as two T$_{RM}$ markers downstream of TGFβ (Runx3 and CD103). As expected, TNF did not induce TGFβ signaling or the expression of Runx3 and CD103 while TGFβ did. However, consistent with the idea that TNF inhibits TGFβ signaling, CD8+ T cells differentiating in the presence of TGFβ were impaired in the phosphorylation of Smad2/3 and the expression of Runx3 or CD103 when TNF was also present (Fig. 5e–h).

To confirm whether the ability of TNF to inhibit TGFβ signaling was NFkB dependent, we repeated the same experiments with DN-IKK2$^{ON}$ CD8+ T cells and used DOX to inhibit NFkB signaling. Confirming our hypothesis, DN-IKK2$^{ON}$ CD8+ T cells remained unresponsive to the effects of TNF on TGFβ signaling (Fig. 5g, i). Thus, TNF inhibits TGFβ signaling and the expression of proteins crucial for CD8+ T$_{RM}$ via NFkB. We also assessed whether TNF and NFkB signaling would affect the induction of other transcription factors important for lung CD8+ T$_{RM}$. Blimp1 was not affected by TNF and/or TGFβ (Fig. 5i). However, T-bet and Eomes were (Fig. 5i). Remarkably, Eomes expression was inhibited by TGFβ[65] and neither TNF nor inhibiting NFkB signaling could revert it (Fig. 5i).

TNF has been shown to be involved in the acquisition of the CD8+ T$_{RM}$ phenotype in in vitro studies[25,66] before, but how changes in TNF levels affect the generation of CD8+ T$_{RM}$ in vivo has not been investigated yet. We thus, decided to test the impact that taming TNF levels during influenza infection would have in the generation of antigen-specific lung CD8+ T$_{RM}$. For this, we used an OT-I adoptive transfer model where we could evaluate the generation of OVA antigen-specific OT-I T$_{RM}$ donors in the lung upon PR8-OVA infection. Based on the data shown in Fig. 5e–i, we predicted that a blockade of TNF would improve the generation of lung CD8+ T$_{RM}$ in high-inflammatory conditions. High inflammation (high TNF) provided by antigen non-specific IAV infection[64,67], caused a decrease in lung OT-I CD8+ T$_{RM}$ (treated with IgG rat isotype) compared with controls. However, blockade of TNF led to a recovery of OT-I T$_{RM}$ numbers and a significant increase over Ig control (Fig. 5j–l). TNF blockade also resulted in an increase in circulating OT-I donors in the lung (Fig. 5l, Supplementary Fig. 8), suggesting that reducing TNF signals may increase CD8+ T$_{RM}$ either by enhancing antigen-specific T cell recruitment to the lung and/or survival of both circulatory and CD8+ T$_{RM}$ cells. Importantly, blockade of TNF signals recovered TGFβ signaling and resulted in an Eomes and Runx3 expression profile that was conducive to CD8+ T$_{RM}$ differentiation (Fig. 5m). Altogether, our data show that TNF-

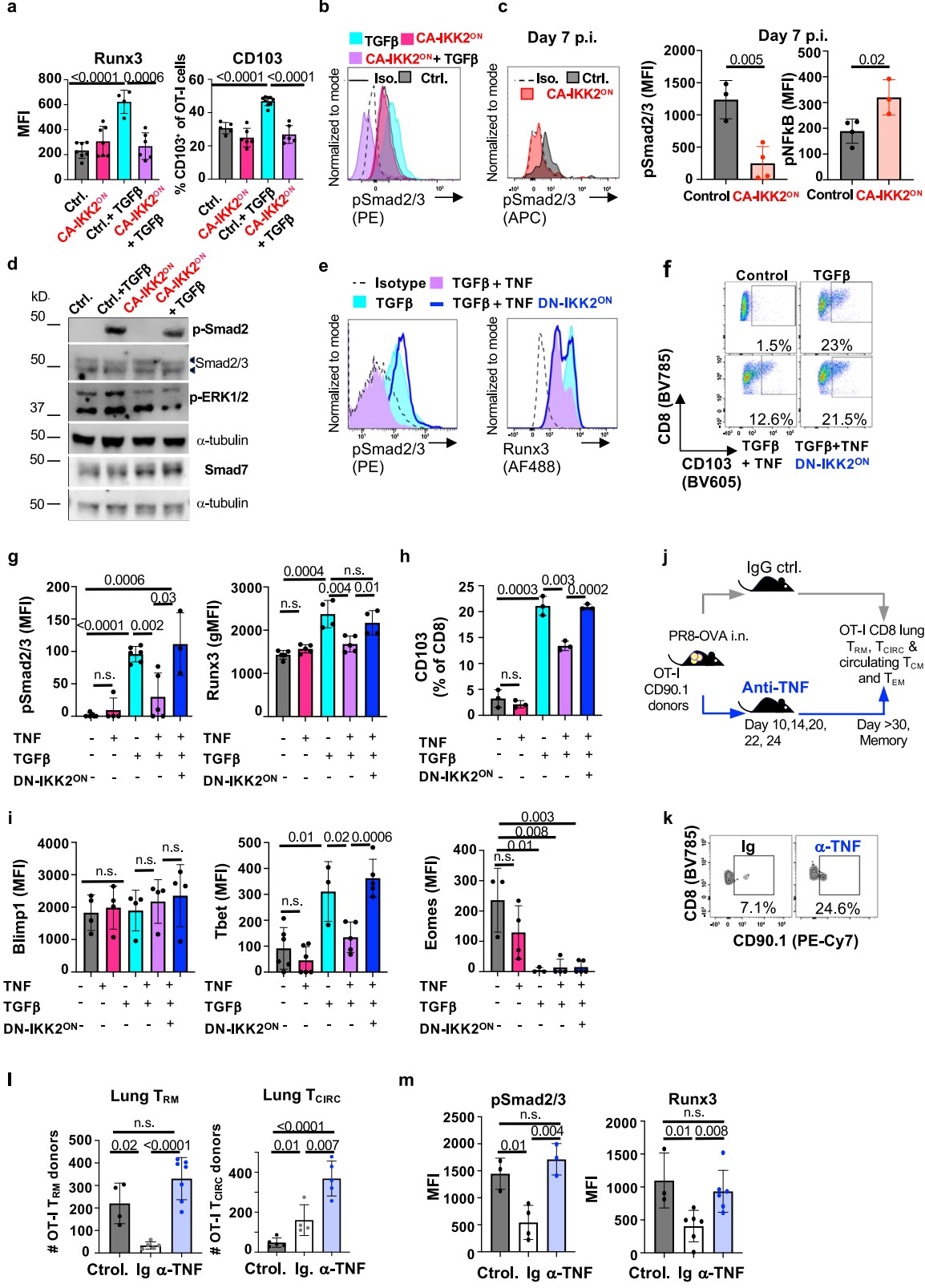

dependent NFkB signals inhibit CD8[+] T$_{RM}$ differentiation via TGFβ signaling and that blockade of TNF can improve CD8[+] T$_{RM}$ generation in conditions of high or continuous inflammation.

**At memory, NFkB signaling promotes lung CD8[+] T$_{RM}$ survival**
Upon influenza infection, lung CD8[+] T$_{RM}$ cells fail to persist, weakening protective immunity against the same or other influenza variants[18,67]. In

prior work using NFkB pharmaceutical inhibitors, we had found that once memory CD8[+] T cells are generated their maintenance depended on NFkB signals[42]. Although these studies did not distinguish between the different T cell memory subsets, the data conflicted with our findings that enhanced NFkB signaling is detrimental to CD8[+] T$_{RM}$ differentiation (Figs. 1, 2, 4). Therefore, we decided to address the impact of NFkB signaling on CD8[+] T$_{RM}$ at memory once memory T cells

**Fig. 5 | TNF and NFkB signaling inhibit TGFβ signaling and downstream $T_{RM}$ markers, CD103 and Runx3. a** Splenocytes from OT-I (control) or OT-IxIKK2$^{fl/fl}$ xGzB$^{Cre}$ mice (CA-IKK2$^{ON}$) were stimulated for 48 h. with OVA and TGFβ. Runx3 and CD103 levels at 24-h. Runx3: $n = 7$ (ctrl. and CA-IKK2$^{ON}$), $n = 4$ (ctrl.+TGFβ), $n = 6$ (CA-IKK2$^{ON}$ + TGFβ). CD103: $n = 5$ (ctrl. and CA-IKK2$^{ON}$), $n = 9$ (ctrl.+TGFβ); $n = 6$ (ctrl. and CA-IKK2$^{ON}$ + TGFβ) mice from 2 (Runx3) and 3 (CD103) independent experiments. **b** Representative histograms showing pSmad2/3 levels upon TGFβ stimulation (30 min). **c** Congenic mice received $10^5$ OT-I or OT-IxIKK2$^{fl/fl}$xGzB$^{Cre}$ donors, followed by PR8-OVA infection (100 pfu). pSmad2/3 and p-NFkB levels were assessed in OT-I $T_{RM}$ cells. For pSmad2/3: $n = 3$, control; $n = 4$, CA-IKK2$^{ON}$ and for pNFkB: $n = 4$, control; $n = 3$ CA-IKK2$^{ON}$ mice from 2 independent experiments. **d** p-Smad2 and p-ERK1/2 or Smad7 (αtubulin = loading control) expression after 30 min. of stimulation. **e–h** Splenocytes from DN-IKK2$^{ON}$ and control mice were stimulated with anti-CD3/CD28. 1 day later, cells were treated ±TNF and ±DOX for 24 h. followed by 30 min. of TGFβ stimulation. Representative histograms (**e**) for pSmad2/3

and Runx3 levels on CD8$^+$ cells or dot plots (**f**) of CD103$^+$ CD8$^+$ cells. **g–i** pSmad2/3, Runx3, Blimp-1, Tbet and Eomes levels and % of CD103$^+$ CD8$^+$ cells determined 24 h. post-TGFβ stimulation. Combined data from left to right graphs: pSmad2/3 $n = 6, 4, 6, 5, 3$; Runx3 $n = 5, 5, 4, 5, 4$; CD103 $n = 3$; Blimp1 $n = 4$, Tbet $n = 6,6,3,5,5$; Eomes $n = 3, 4, 3, 4, 5$ mice pooled from (**g**) or representative (**h**) of 2 independent experiments. **j** Experimental design. **k** Representative dot plots with % OT-I $T_{RM}$ at 30 d.p.i. **l** OT-I $T_{RM}$ (IV-) and circulating (IV$^+$) cell numbers at 30 d.p.i. $T_{RM}$: $n = 4$, control; $n = 5$, Ig; $n = 7$, anti-TNF. $T_{CIRC}$: $n = 5$, control; $n = 4$, Ig; $n = 5$ anti-TNF mice from 2 independent experiments (**m**) pSmad2/3 and Runx3 levels in $T_{RM}$ from (**l**). For pSmad2/3/Runx3: $n = 3$, control; $n = 4$, Ig; $n = 3$ anti-TNF. $T_{CIRC}$: $n = 3$, control; $n = 6$, Ig; $n = 6$ anti-TNF, representative (pSmad2/3) or pooled (Runx3) from 2 independent experiments. Bars represent mean values ± SD. $P$-values determined by two-tailed unpaired $t$ test, n.s. not significant. Source data provided as a source data file.

had been formed. For this, we used the CA-IKK2$^{ON}$ inducible model and allowed the generation of CD8$^+$ $T_{RM}$ in the lung upon IAV infection. 30 d.p.i, we inhibited NFkB signaling (DOX treatment) and compared control and DOX-treated mice for changes in the number of CD8$^+$ $T_{RM}$ in the lung parenchyma. We observed a ~4-fold increase in the number of IAV-specific CD8$^+$ $T_{RM}$ cells in the lung when NFkB signaling had been induced (Fig. 6a). Strikingly, this was the opposite effect that increasing NFkB signals during contraction had in CD8$^+$ $T_{RM}$ generation (Fig. 1). The increase in CA-IKK2$^{ON}$ CD8$^+$ $T_{RM}$ at memory correlated with higher levels of CD122 and Bcl-2, suggesting NFkB signals at memory mediate CD8$^+$ $T_{RM}$ survival (Fig. 6b).

In most tissues, $T_{RM}$ maintenance is independent of the input of circulating memory T cells ($T_{CIRCM}$)[68,69]. However, in the lung this is still controversial[67,70]. In our studies, increasing NFkB signals at memory significantly boosted the frequency and number of IAV-specific CD8$^+$ $T_{CIRCM}$ and $T_{RM}$ in the lung (Fig. 6c and Supplementary Fig. 9). We performed additional experiments to test whether the increase in CD8$^+$ $T_{RM}$ upon transient induction of NFkB signals was due to an increase in cellular turnover or homeostatic proliferation. For this, we performed BrdU labeling in vivo upon DOX administration[67]. We observed no changes in BrdU incorporation or lung CA-IKK2$^{ON}$ $T_{RM}$ or $T_{CIRC}$ cell proliferation (Supplementary Fig. 10). However, both lung CA-IKK2$^{ON}$ CD8$^+$ $T_{RM}$ and $T_{CIRC}$ exhibited much higher levels of Bcl-2 than their control counterparts further suggesting that NFkB drives persistence of lung CD8$^+$ $T_{RM}$ by boosting T cell memory survival (Supplementary Fig. 10).

Remarkably, inducing NFkB signaling at memory was also beneficial for the CD8$^+$ $T_{CM}$ pool (CD62L$^{hi}$ expressing cells) as we also observed a significant increase in the number of IAV-specific CD8$^+$ $T_{CM}$ in draining lymph nodes after DOX treatment (Fig. 6d, e).

We also tested whether transient induction of IKK2/NFkB signaling at memory would provide a temporary or a long-term increase in IAV-specific CD8$^+$ $T_{RM}$. To test this in a CD8$^+$ T cell-intrinsic manner, we transferred OT-I naïve donors from OT-I.CD90.1xCD2rtTAxCA-IKK2 mice into CD90.2 B6 hosts and infected the mice with PR8-OVA. At memory, we treated chimeric mice with or without DOX for 10 days as in Fig. 6a and then removed treatment in a cohort of the DOX-treated mice and not in the other. Fifteen days after interrupting the induction of NFkB signaling (DOX treatment), we assessed IAV-specific CD8$^+$ $T_{RM}$ decay. While continuous provision of NFkB signals led to a steady increase of lung CD8$^+$ $T_{RM}$, removal of IKK2 constitutive signaling reverted the numbers of CD8$^+$ $T_{RM}$ to control levels (Supplementary Fig. 11). From these, we concluded that an increase in NFkB signaling at memory can boost lung CD8$^+$ $T_{RM}$, as long as the provision of NFkB signaling is continuously maintained. Interruption of NFkB signaling also affected lung circulating memory, most likely because NFkB signaling supports the survival of all CD8$^+$ memory subsets.

Finally, we performed similar experiments using the DN-IKK2$^{ON}$ inducible model. Inhibiting NFkB signaling once CD8$^+$ $T_{RM}$ cells have

formed, had the converse effect, resulting in a loss in CD8$^+$ $T_{RM}$ numbers and a decrease in the expression of survival factors (CD122 and Bcl-2) (Fig. 6f, g). This was similar for the circulatory T cells in the lung and for $T_{CM}$ cells in draining lymph nodes, further demonstrating that NFkB signals are key for the survival and maintenance of CD8$^+$ $T_{CIRCM}$ and $T_{RM}$ upon influenza infection (Fig. 6h–j and Supplementary Fig. 9).

In summary, our data reveal that NFkB signaling differentially affects tissue-resident memory depending on the stage of differentiation of the CD8$^+$ T cell (before and after becoming CD8$^+$ $T_{RM}$). Most importantly, our results support the idea that enhancing NFkB signaling in CD8$^+$ T cells once the $T_{RM}$ pool has been established could improve CD8$^+$ $T_{RM}$ survival in tissue and circulatory memory populations.

## Discussion

T-cell memory in tissues is an essential part of mucosal immunity that protects against infection and can contribute to disease. Here we show that the pro-inflammatory signaling pathway IKK2/NFkB, is critical for both the generation and maintenance of the CD8$^+$ $T_{RM}$ pool after infection. Our results mainly refer to influenza-specific CD8$^+$ $T_{RM}$ in the lung, a tissue where maintaining a long-lived CD8$^+$ $T_{RM}$ pool is crucial for protective immunity[18] but challenging, due to the CD8$^+$ $T_{RM}$ short life-span[71]. The reasons why resident memory CD8$^+$ T cells are short-lived in the lung but not in other tissues are still unclear. However, our work suggests that boosting NFkB signaling at the end of the immune response might offer a therapeutic opportunity to increase CD8$^+$ $T_{RM}$ survival and protective immunity upon infection or vaccination. Furthermore, since improved survival also occurred for CD8$^+$ $T_{CM}$ of the draining lymph nodes, this suggests a controlled increase in NFkB signaling could boost several subsets of the T cell memory pool.

It is striking that the same signaling pathway, NFkB, operates in opposite manners for CD8$^+$ T cells depending on their differentiation stage (during $T_{RM}$ differentiation and after). This could be due to epigenetic modifications that regulate the accessibility of NFkB to specific gene loci depending on the differentiation stage of a CD8$^+$ T cell. Alternatively, changes in the environmental cues as the infection resolves could also explain the differential impact on CD8$^+$ $T_{RM}$ when levels of NFkB signaling increase. Although our findings cannot distinguish between these two possibilities, our data suggest that NFkB signaling does interfere with TGFβ to skew CD8$^+$ T effectors away from $T_{RM}$. In this regard, it is important to note that TGFβ levels in tissue can change depending on age and in the context of diseases such as infection, autoimmunity, asthma, or fibrosis[72], although how these conditions affect CD8$^+$ $T_{RM}$ generation or survival is still poorly understood.

Multiple signals can locally trigger the induction of NFkB signaling in CD8$^+$ T cells, including antigen, TLRs and pro-inflammatory cytokines[35–37]. We show that TNF, a known driver of NFkB[35] is able to

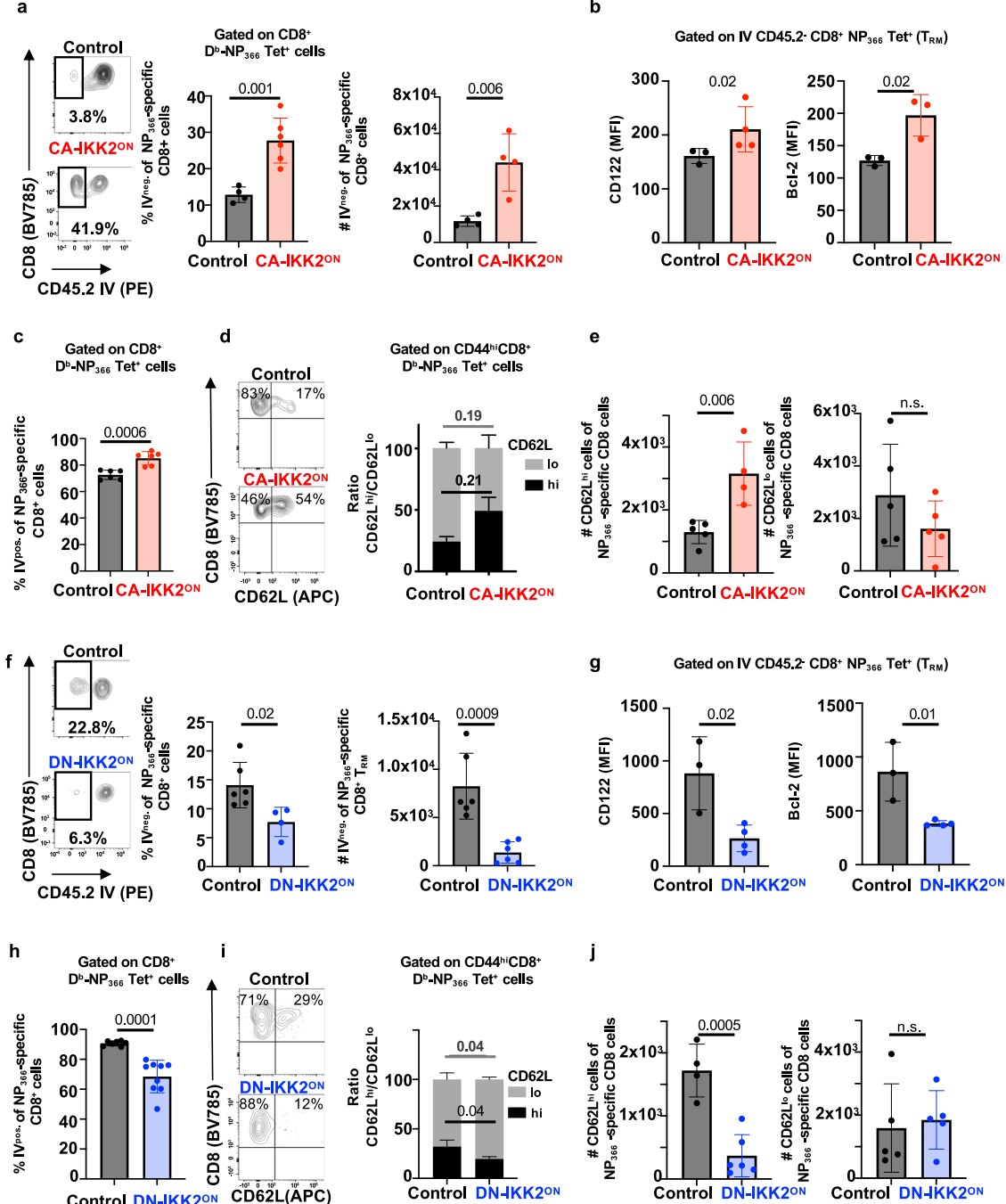

**Fig. 6 | Increasing NFkB signaling at memory improves CD8⁺ T$_{RM}$ survival.**
Groups of control or CD2rtTA x CA-IKK2 mice (**a–e**) or CD2rtTAxDN-IKK2 mice (**f–j**) were infected with influenza X31 (1000 pfu). At 30 d.p.i. mice were fed a ±DOX (CA-IKK2$^{ON}$) diet for 15 days. (**a**, **f**). Mice were infected with Influenza X31 and T$_{RM}$ were identified by IV neg. labeling. Representative dot plots showing % of IV neg. among influenza-specific CD8⁺ T cells. Graphs show frequencies and numbers of NP$_{366–374}$ specific CD8⁺ T$_{RM}$ cells. Combined data for %, # in (**a**): $n = 4,4$ (control), $n = 6,4$ (CA-IKK2$^{ON}$) $n = 6,6$ and for %, # in (**f**): $n = 6, 6$ (control), $n = 4,6$ (DN-IKK2$^{ON}$) mice pooled from 2 independent experiments (**b**, **g**) Bcl-2 and CD122 expression in influenza-specific CD8⁺ T$_{RM}$ cells in the lung parenchyma. Graphs show combined data for CD122, Bcl-2 in (**b**): $n = 3,3$ (control); $n = 4,3$ (CA-IKK2$^{ON}$) and in (**g**): $n = 3.3$ (control), $n = 4,4$ (DN-IKK2$^{ON}$) mice, representative of 2 independent experiments. (**c**, **h**)

Frequencies of circulating (IV⁺), NP$_{366}$ specific CD8⁺ T cells in the lung. Graph shows combined data for (**c**): $n = 6$ (control and CA-IKK2$^{ON}$), for (**h**) $n = 7$ (control) and $n = 9$ (DN-IKK2$^{ON}$) mice pooled from 2 independent experiments. **d**, **e** and **i**, **j** Representative dot plots showing % of CD8⁺ CD62L$^{hi/lo}$ cells among influenza NP$_{366-374}$ CD8⁺ T$_{MEM}$ in mediastinal lymph nodes. Frequencies and numbers of influenza NP$_{366–374}$-specific (CD8⁺, Db-NP-tet⁺, CD44$^{hi}$, CD62L$^{hi}$) and (CD8⁺ Db-NP-tet⁺ CD44$^{hi}$ CD62L$^{lo}$) memory subsets were determined by flow cytometry. Combined data for CD62L$^{hi}$, /CD62L$^{lo}$ in (**e**): $n = 5/5$ (control); n = 4/5 (CA-IKK2$^{ON}$) and for CD62L$^{hi}$/ CD62L$^{lo}$ in (**j**): $n = 4/5$ (control); $n = 6/5$ (DN-IKK2$^{ON}$) mice pooled from 2 independent experiments. Bars represent mean values ± SD. P-values were determined by two-tailed unpaired t test, n.s. not significant. Source data provided as a source data file.

inhibit TGFβ dependent signaling and T$_{RM}$ programming in CD8⁺ T cells. Furthermore, our data suggest that blocking TNF late in the response can increase the generation of CD8⁺ T$_{RM}$. This role of TNF is different from the one described by previous studies from Harty's group at memory, when blocking TNF resulted in a reduction in CD8⁺ T$_{RM}$[67,70]. Their results, however, are consistent with our data at memory supporting the idea that NFkB signaling effects on CD8⁺ T$_{RM}$ differentiation, before and after CD8⁺ T cell memory is established, are

different. TNF and other pro-inflammatory cytokines such as IL-6 are heavily produced in pathological settings of chronic inflammation and could easily affect the levels of NFkB signals[73–75] that a CD8+ T cell experiences in tissue. For the CD8+ T cells that also encounter TGFβ locally, this could decrease their likelihood to become CD8+ $T_{RM}$. Further evaluation of the dynamics of CD8+ $T_{RM}$ during infections with a strong pro-inflammatory profile could provide insight into how overt inflammation may affect long-term immunity and inform of specific therapeutics targeting NFkB or its pro-inflammatory drivers to either boost or deplete tissue-resident memory. This might be especially relevant in the context of immune treatments that are linked to high levels of inflammation and that in the case of cancer[76,77] or auto-immunity (such as rheumatoid arthritis) have a $T_{RM}$ component that affects disease outcome[14,78]. In the same line, it would also be important to evaluate how current treatments targeting TNF or NFkB signaling in the clinic, affect the establishment of tissue-resident memory in patients.

NFkB signaling did not affect CD8+ $T_{RM}$ in the same manner as it did for other $T_{CIRCM}$ subsets in the mediastinal lymph nodes, indicating that NFkB signaling is a key regulator of T cell memory diversity. Modulation of NFkB signaling levels may serve as an opportunity to regulate specific T cell memory subsets depending on their role in disease.

Finally, our findings also underscore the impact that fluctuating levels of a single signaling pathway can have on the quality of T cell memory depending on how and when during the infection these levels change. This may be particularly important for pleiotropic signaling pathways such as NFkB where multiple stimuli feed in (including patient's treatments) and can easily add up to shape T cell fate. Incorporating this concept into current vaccine strategies could aid to improve their long-term efficacy.

## Methods

All procedures were conducted according to the NIH guidelines for the care and use of laboratory animals and biological safety and were approved by the University of Missouri Institutional Animal Care and use and Institutional Biosafety Committees.

### Mice

OT-I.CD90.1 (or Thy1.1+) TCR transgenic strain were generated in the lab by crossing OT-I mice (C57BL/6-Tg (TcraTcrb)1100Mjb/J 003831 from Jackson Laboratory) and B6.CD90.1 mice (B6.PL-$Thy1^a$/CyJ Strain no. 000406 from Jackson Laboratory); C57BL/6J (B6.CD90.2), B6.SJL-$Ptprc^a$ $Pepc^b$/BoyJ (CD45.1 congenic C57BL/6), B6.Cg-$Gt(ROSA)26Sor^{tm4(Ikbkb)Rsky}$/J (IKK2-CA$^{fl/fl}$), B6 -Tg (GzB-cre)1Jcb/J (GzB-Cre) mice were from Jackson Laboratory(Bar Harbor, ME). OT-I.CD90.1XIKK2CA$^{fl/fl}$xGzB$^{Cre}$; CD2rtTA x CA-IKK2 (tetracycline-inducible constitutive active IKK2) and CD2rtTA x DN-IKK2 (tetracycline-inducible dominant negative IKK2) mice were maintained under specific pathogen-free conditions at the University of Missouri. All mouse strains are on the C57BL/6 background and were screened for transgene homozygosity by PCR. Mice were aged between 8 and 13 weeks at the time of infection for all experiments. Female and male mice were not disaggregated as no gender bias was found. Experiments in Fig. 3 required designated genders otherwise, experiments were gender matched. Infection and maintenance of mice infected with influenza virus or vesicular stomatitis virus occurred in an ABSL2 facility at the University of Missouri. Mice were euthanized by $CO_2$ inhalation followed by cervical dislocation. Mouse housing conditions: Light/dark cycle: 12/12 with lights on at 7 AM and off at 7 PM. Ambient temperature: 70–71 °C. Humidity: 50–55%. All animal procedures were conducted according to the NIH guidelines for the care and use of laboratory animals and were approved by the University of Missouri Institutional Animal Care and use Committee.

### Reagents, antibodies, and primers

Provided in Supplementary Tables 1–3 in supplementary information.

### Virus infections

Mice were infected intranasally with 1000 pfu influenza A/HKx-31 (X31, H3N2) for sublethal infection or intravenously with vesicular stomatitis virus (VSV) ($2 × 10^6$ pfu), unless otherwise indicated in Figure legends. For heterologous infection experiments, mice were primed intranasally with $5 × 10^4$ pfu VSV-OVA, then challenged 30 days later with 5000 pfu of influenza A/PR8-OVA (PR8, H1N1).

### Viral titers

The $TCID_{50}$ of influenza virus was determined using MDCK cells as described (74). Briefly, lung samples were homogenized using a Mini-BeadBeater (BioSpec, Bartlesville, OK) and cleared homogenate was used to inoculate confluent MDCK cell monolayers. 24 h post-inoculation, the supernatant was discarded and replaced with fresh media (DMEM containing 0.0002% Trypsin). Agglutination of chicken RBCs (Rockland Immunochemicals Inc., Limerick, PA) was utilized to determine the presence of influenza virus after 3 days of culture.

### In vivo antibody labeling and flow cytometry

For in vivo antibody labeling and differentiation of T cells circulating in the vasculature or resident in parenchyma ($T_{RM}$) tissues, three minutes before being killed, mice were injected intravenously via tail vein injection with 2 µg PE-labeled CD45.2 (clone 104, Biolegend) or PE-labeled CD8β, (clone Ly-3, BD Biosciences). Lungs, kidney, spleen, and mediastinal lymph node tissues were harvested, and lymphocytes isolated. Next, lymphocytes were stained in vitro with anti-CD8α antibodies along with fluorochrome-conjugated antibodies specific of other surface markers resuspended in FACS buffer (PBS/1% fetal bovine serum. For immunostaining of intracellular markers such as transcription factors T-bet, Eomes or Nur77, after staining of cell surface markers, cell samples were rinsed, fixed and permeabilized with Cytofix/cytoperm. Stained cells were run on a LSR Fortessa flow cytometer (BD, San Jose, CA, data collected with FACS Diva 6.0 by BD) or run on a Cytek Aurora spectral flow cytometer using the Cytek Spectraflo 3.0.3 software. Flow cytometry data was analyzed using with FlowJo software version 10.4 (Tree Star, Inc., Ashland, OR). It should be considered that our observations refer mainly to the generation and maintenance of influenza-specific memory CD8+ T cells in the lung parenchyma identified through IV labeling, a method widely used to study $T_{RM}$ cells in the field. One limitation of this approach is that it can only provide a snapshot of the cells that are residents in tissue at a given time and it is less accurate at quantifying T cells in transition. A table of antibodies is provided in supplementary information.

### Intracellular cytokine staining

Lymphocytes were isolated from the lungs of VSV-OVA-challenged mice and stimulated ex vivo with OVA peptide 1 µM) in the presence of Golgi-Plug (BD Biosciences) for 5 h. Following incubation, cells were harvested and antigen-specific CD8+ T cells were assessed for the expression of TNF and IFNγ by flow cytometry as in[44].

### Intracellular phosphorylated-SMAD2/3 and phosphorylated-NFkB detection by immunostaining for flow cytometry

Cell suspensions from lung, lymph nodes or spleen were fixed with 4% Formaldehyde aqueous solution for 20 min at room temperature, followed by two rinses with PBS/1% fetal bovine serum solution (FACs buffer). Permeabilization was performed with ice-cold methanol at −20 °C overnight. The next day, cells were stained with anti-phospho SMAD2/3 or anti-phospho NFkB in PBS/1% fetal bovine serum solution. For Phospho NFkB staining, inhibitors of phosphatases (sodium pervanadate 1 mM, and sodium fluoride 100 mM) were added at the same time.

## Genotyping

Genomic DNA from mouse tail clippings was isolated using the AccuStart II kit. The genomic DNA was amplified using rtTA forward and rtTA reverse primers to identify the Cd2rtTA transgene and IKK2 forward and IKK2 reverse primers were used to identify the Ikbkb mutant transgene (constitutive active form). PCR reactions were performed in an Eppendorf Mastercycler according to refs. 40,41,79. PCR products were resolved with a 1% agarose gel. Primer sequences are provided in supplementary information (Supplementary Table 2).

## CD8$^+$ T cell enrichment and adoptive transfer

Splenocytes were harvested from CD2rtTA x CA-IKK2 mice and poly-clonal CD8$^+$ T cells were purified by magnetic selection (CD8$^+$ T cell isolation kit by Miltenyi Biotech Auburn, CA). $5 \times 10^5$ CD8$^+$ polyclonal or $10^4$ OT-I.CD90.1XIKK2CA$^{fl/fl}$xGzB$^{Cre}$ monoclonal naïve T cells were adoptively transferred into congenic C57BL/6 mice 1 day prior to intranasal infection with influenza or VSV-OVA virus, respectively.

## BrdU Labeling

BrdU was obtained from Biolegend. Mice were given a bolus dose of BrdU (2 mg I.P.) and given a supplemental dose (0.8 mg daily) of BrdU in their drinking water for 6 additional days. BrdU incorporation of lymphocytes was determined by flow cytometry. Lymphocytes were isolated and subjected to appropriate phenotyping antibodies. Cells were fixed in 4% formaldehyde then permeabilized with 0.5% Triton followed by DNase treatment (Sigma) then stained with AlexaFluor 647 conjugated anti-BrdU antibody (clone 3D4, Biolegend).

## TNF Blockade during influenza infection

Host Mice received $1 \times 10^5$ congenic OT-I CD 90.1 T cells followed by infection with PR8-OVA (100 pfu). Groups were treated with either Anti-TNF blocking antibody or Rat IgG control. Specifically, TNF signaling was blocked by treating mice with an Ultra-LEAF purified, TNF-specific antibody (clone MP6-XT22) while an Rat IgG1κ isotype control was used in parallel in control treatment mice .100 µg of antibody was administered via intranasal instillation on days 10, 14, 20, 22, and 24 post-infection. Then, treated mice were reinfected with influenza PR8 (100 pfu) at 10 d.p.i. and compared with non-treated and non-PR8 re-infected controls for numbers of OT-I $T_{RM}$ and $T_{CIRC}$ in the lung as well as for their levels of pSmad2.3 and RunX3.

## In vitro culture and cytokine stimulation

Splenocytes isolated from OT-I.CD901.1XIKK2CA$^{fl/fl}$ xGzB$^{Cre}$ mice were stimulated with 20 nM OVA peptide for 48 h at a concentration of $1 \times 10^6$ cells/ml. TGFβ (R&D Systems, Minneapolis, MN) was then added to a final concentration of 50 ng/ml. At 30 min and 24 h post TGFβ stimulation, cells were harvested for analysis by flow cytometry and western blotting. Splenocytes from CD2rtTA x DN-IKK2 mice were stimulated in vitro at $1 \times 10^6$ cells/ml with 10 µg/ml anti-CD3 (clone 145-2C11) and 10 µg/ml anti-CD28 (clone 37.51) (ThermoFisher Scientific, Waltham, MA). Following 24 h of stimulation, cells were divided and incubated in the presence or absence of 125 ng/ml recombinant TNF (R&D Systems, Minneapolis, MN) for 24 additional hours. The cells were again divided and incubated in the presence or absence of 50 ng/ml TGFβ (R&D Systems, Minneapolis, MN). Cells were harvested at 30 min and 24 h post addition of TGFβ and analyzed by flow cytometry.

## Western blotting

In vitro stimulated cells were lysed in lysis buffer containing 10 mM HEPES, 10 mM KCl, 0.1 mM EDTA, 0.2 mM EGTA, 0.5% NP40, 1 mM DTT, 2 mM Na$_3$VO$_4$, 20 mM NaF, 1 mg/ml Leupeptin, 1 mg/ml Aprotinin, 1 mM PMSF. Samples were resolved on a 10% SDS-PAGE gel and transferred to nitrocellulose membrane. Membranes were blocked with Blotting Grade Blocker (Bio-Rad, Hercules, CA) and probed with specific primary and secondary antibodies. Blots were imaged on a Li-Cor Odyssey XF (Li-Cor, Lincoln, NE) and analyzed using Image Studio v5.2.5 (Li-Cor, Lincoln, NE). A table of antibodies is provided in supplementary information. Uncropped, unprocessed scans of the most important blots appeared in supplementary information Figs. 13, 14

## Statistical analysis

Statistical analysis was performed using the Prism software (GraphPad v. 9.4.1). Data are presented as mean ± standard deviation. Statistical significance to compare two quantitative groups was evaluated using a two-tailed $t$ test. $P$ exact values are indicated in each of the figure legends.

Raw data related to the studies in this article are available in OSF (see data availability section). A Source data file is provided with this article.

## Reporting summary

Further information on research design is available in the Nature Portfolio Reporting Summary linked to this article.

## Data availability

Raw data generated in this study have been deposited in the public Open Science Framework (OSF) https://osf.io/2vtur/, with no accession code needed. Raw flow files are available upon request due to size restrictions. Access can be obtained by contacting the corresponding author. Uncropped, unprocessed scans of the blots appeared in supplementary information Figs. 13, 14. All other data are available in the article and its Supplementary files or from the corresponding author upon request. Source data are provided with this paper.

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

## Acknowledgements

The authors thank the University of Missouri English Department for grammar edits. We thank Dr. Nick Goplen, Dr. Vikas Saxena and Bianca Gordon for discussion and support optimizing the tests for validation of the inducible mouse models; Elida Lopez for screening and help with sample processing. We thank Dr. Rose Zamoyska, Dr. Bruce Richardson, and Faith Strickland for generously providing us with CD2rtTA mice; We thank Dr. Bernd Baumann and Dr. Thomas Wirth for generously providing us with IKK2-CA and IKK2-DN mouse strains. MU Office of Research and School of Medicine animal vivarium staff for assistance with mice. Kathy Schreiber, Daniel Jackson, Yue Guan and Susan Rottinghaus for assistance in the Flow Cytometry Core Facility. Dana Weir-Guffey, Landon MacDowell, Angela Hall, Raye Lynn Allen in the Office of Animal Research and in the School of Medicine and Bond Life Science Center Animal Vivaria respectively for extraordinary support. Supported by National Institutes of Health grants R01 AI110420-01A1 and R56 AI110420-06A1, 1U01CA244314 as well as internal funding from the School of Medicine (ET); NCI CA244314 (ET), NIH IB026560 (MAD), NIH-REACH (MAD); T32 GM135744(DL).

## Author contributions

C.J.P., D.L., and K.M.K. performed experiments; C.J.P., D.L., and E.T. analyzed the data; K.M.K. and M.J.Q. generated and validated mice; C.J.P., M.A.D., and E.T. made important conceptual contributions; and C.J.P. and E.T. designed the experiments. E.T. and M.A.D. wrote the manuscript. M.A.D. and C.J.P. edited the manuscript. M.J.C. edited the grammar of the manuscript.

## Competing interests

The authors declare no competing interest.
