## [Peer Review File · Nature Communications]

IKK2/NFkB signaling controls lung resident CD8 T cell memory during influenza infectionREVIEWER COMMENTS

Reviewer #1 Trm, T cells, virus infection (Remarks to the Author):

An inducible tetOn expression system has been used to determine how temporal changes in IKK2/NFkB signaling influence survival of TRM cells in the lungs during infection with influenza A virus and VSV. Constitutively-active and dominant-negative forms of IKK are expressed in CD2+ cells. The authors report that the level of NFkB signaling inversely correlates with the numbers of pathogen-specific TRM cells in the lungs. The study further shows that doxycycline-dependent induction of IKK2 results in upregulation of NFkB dependent genes CD69 and Eomes. Although novel technology is used to make several interesting observations, the data are not very robust. Enthusiasm of the study is reduced by an over-reliance on IV staining to identify TRM cells. No figures show TRM cells collected from the lungs by bronchoalveolar lavage. Several figures show very low frequencies of TRM cells, that may be insufficient for reliable statistical analyses. Frequent changes between infection models makes data interpretation challenging. The rationale for some experiments is unclear.

1) Line 90-93. That authors indicate that NFkB signaling was manipulated “after the peak of the T cell response” and “during the contraction phase”. A relevant figure is not indicated. Does this simply mean continuous treatment between 5-30dpi? This question arises again on line 123. Line 105 states that 5 dpi is peak of the CTL response to IAV infection. According to most published studies, the CTL response peaks ~9dpi.

2) Some figures show very low frequencies of Tetramer+ TRM cells (less than 0.1%). Figure 1 should include flow plots show tetramer staining in the mutant mice plus/minus dox, as well as the numbers of PA-specific TRM cells in the lungs.

3) Line 115. The authors indicate that IAV specific CD8 TRM cells were found in spleen 30dpi. However, these cells are incompletely characterized. Although IV staining can distinguish cells in the red and white pulp, this technique only provides a snap-shot view of the CTL response and does not provide evidence of prolonged residence in a specific tissue. There is concern that the staining was not 100% efficient and that some circulating cells were not stained by the injected antibody. To conclude that these splenocytes are indeed TRM cells, the authors should analyze canonical markers of TRM cells (CD69, CD103, Hobit) and study migration by parabiosis.

- 4) Since CD2 is expressed on NK cells and DCs, the CTL response may have been altered by off-target effects on other types of cells. Since inflammation impacts T cell-migration/survival, authors should verify whether prolonged Dox treatment causes immune pathology and inflammation.
- 5) Line 119. The authors state that they “did not observe important defects in the lung IAV-specific CD4 T cell memory compartment”. No functional or phenotypic data are provided. Figure 2 does not clearly indicate how many mice were analyzed (six?), however an apparent reduction in the numbers of CD4 T cells (Fig 2C) may be significant if larger cohorts are used.
- 6) Line 122. The authors indicate that they used an adoptive transfer model to confirm whether the effect of NFkB signaling was CD8 T cell intrinsic. Figure 3 shows that increased NFkB signaling augments T cell effector function (Fig 3e). Since proinflammatory cytokines can indirectly induce T cell death by activating myeloid cells in the local tissues, the loss of TRM cells may be due to extrinsic inflammation.
- 7) Line 133. The authors state that a published approach was used to deplete circulating CD8 T cells, while sparing CD8 TRM. No reference is provided. As male mice are generally larger than females, there is concern that gender might influence disease progression.
- 8) Figure 4. It is unclear why CD69 and CD103 were analyzed separately. Most mucosal TRM cells express both markers, as indicated on line 50.
- 9) The authors should discuss the relevance of Nur77 expression. Does this marker indicate a response to persisting antigen?
- 10) The manuscript includes some typographical errors.

While I consider this paper to be quite interesting, some major revisions are required before publication.

Sincerely, Linda S. Cauley

Reviewer #2 T cells (Remarks to the Author):

The authors examine the impact of enhancing and inhibiting NFκB signaling on generation and maintenance of distinct memory CD8⁺ T cell subsets. Using constitutively active (CA) and dominant negative (DN) IKK2 constructs under a tetracycline inducible genetic system (or, in some experiments, Cre-mediated induction), the authors find a series of interesting findings. First, CA IKK2 induction following the peak expansion of the CD8⁺ T cell response promotes generation of recirculating “central” memory CD8⁺ T cells yet inhibits generation of “resident” memory CD8⁺ T cells (TRM) in the lungs (influenza infection) and kidney (VSV infection). This includes dysregulated expression of several genes associated with TRM, including reduced CD103, Runx3, Nur77 and CD122, but increased Eomes expression. Expression of DN IKK leads to reciprocal changes for the most part. The authors show that signals through TGF-β (which induce CD103 and are key to establishing TRM) are antagonized by NFκB stimulating factors such as TNFα. Interestingly, the authors find that the timing of NFκB regulation critically affects outcomes – induction of CA IKK2 after CD8⁺ T cell memory has already formed leads to improved (not reduced) persistence of TRM, indicating that the roles of NFκB in control of TRM change during generation and maintenance.

The authors have developed elegant and powerful mouse models to dissect the impact of NFκB signaling on the fate and homeostasis of CD8⁺ T cell memory populations, building on their previous studies showing a role for NFκB signals in memory T cell preservation. While the findings are complicated (reflecting the underlying biology), the message is clear, the data compelling and the studies important for understanding the role of the NFκB pathway in shaping T cell memory. In particular, the authors findings suggest a potential role for diminished NFκB signaling as a cause of the accelerated decay in the lung TRM population (compared to TRM in other tissues), which is well known but poorly understood.

There are some concerns that will need to be addressed, however.

1) The authors focus on the IKK2 CA model, with less data on the IKK2 DN transgenic mice. While there is no need to repeat all studies in both models, it would have been powerful for

the authors to show the impact of cell-intrinsic effects of IKK2 DN induction in generation of TRM (essentially following the approach outlined in Fig. 2e). This would allow the authors to determine whether effects of both increased and decreased NFkB are CD8+ T cell-intrinsic.

2) Along the same lines, it would be interesting to see whether induction of IKK2 DN in the established memory CD8+ T cells would lead to accelerated loss of TRM (i.e. use of the DN model as in Fig. 6). This reviewer appreciates, that such studies would take considerable time, however, but some consideration of this issue would be valuable.

3) Do the authors have any data on whether enhanced maintenance of lung TRM by late IKK2 CA induction may reflect increased proliferation (in addition to the proposed Bcl-2 related avoidance of cell death)? BrdU incorporation assays could be used to assess this.

4) The authors carefully describe the loss of parenchymal (iv antibody -ve) cells in the lung, and also show reduced numbers of TRM-phenotype cells when IKK2 CA is induced.

However, it is less clear whether there is a change in the numbers of recirculating cells in the lung parenchyma by IKK2 CA (or DN) expression – i.e. whether there is an increase or decrease in the tissue-trafficking “effector” memory population. This could be shown as the frequency and numbers of CD69-ve antigen-specific cells in the iv antibody -ve pool (perhaps these data could be added to Fig. 4c).

5) Have the authors any data on whether transient induction of IKK2 CA during memory maintenance (as in Fig 6) leads to sustained increase in the lung TRM pool? Or, instead, does that population “crash” once IKK2 CA expression is allowed to decline? This relevant as to whether the provision of increased NFkB signaling “rescues” lung TRM maintenance or only provides a temporary reprieve in survival.

Reviewer #3 NFkb, IKK (Remarks to the Author):

Summary of the paper

This manuscript describes new observations of lung residential CD8 memory T cells in transgenic mice that express a constitutively active form and a dominant-negative form of I κ B kinase B (IKKB). The authors infect these mice with influenza A virus or VSV expressing OVA and measure the level of antigen reactive CD8 T cells in the lung tissue using tetramers. When they induce a constitutively active form of IKKB from 5 days after infection till day 30, they observe a significant reduction in tetramer-positive lung residential CD8 T cells. Conversely, they observe an increase in tetramer+ CD8 Trm cells in the lung when a dominant negative form is expressed. To understand the mechanism of these observations, they further analyze the cells that remain in the CA-IKKB transgenic mouse and found that the frequency and cell numbers of the tetramer+ cells expressing CD69 and CD103 are reduced in these mice. Using an in vitro system, the authors show that TGF-beta signaling is inhibited by the active form of IKK B and TNF, suggesting that TNF may be a potential factor that can activate NF- κ B during inflammation, blocking TGFbeta signaling, and causing a reduction in CD8 lung residential memory T cells.

Comments

The most noteworthy result from this work is shown in Figure 1 and Figure 6 whereby expression of the active form of IKKB have opposite effect on CD8 lung Trms depending on the timing of expression. Contrary to the day 5-30 induction of CA IKKB, post day 30 expression of CA IKKB increases the number of tetramer-positive T cells.

While these observations are novel and potentially significant for the field, the current manuscript does not provide sufficient results to support their claim. Followings are the major issues that need to be addressed.

- 1) In Figure 6, the authors show that activation of NF- κ B 30 days after infection enhances the CD8 TRM. They claim this suggests that the Nf- κ B signal supports the expansion of Trm

in the late phase of infection. This is a puzzling observation because the experiment is designed such that no antigen stimulation is presumably present. If so, why only tetramer-positive Trm cells expand? There are other CD8Trm exist in the lung. The data show over 99% of Trm cells is tetramer negative. The induction of CA IKKB is driven by CD2, thus all T cells are expressing active IKKB. Why the other Trm cells do not expand in response to the induction of CA IKK? This point requires rigorous clarification.

2) In supplementary Figure 2, they show that expression of CA IKKB does not induce caspase 3 activation or FasL expression (surface stain), and conclude that this transgene expression does not induce cell death. However, there is no in vivo data that support this assumption. This manuscript needs to determine if constitutive and sustained IKKB active form expression is causing accelerated cell death, migration (discussed below), or blocking differentiation as they claim. It has been shown at TGF-beta not only induces differentiation but also inhibits cell death by many systems.

3) In Fig.1b, the data show that Tcm in the mediastinal lymph nodes in CA KKK mouse significantly increases while Tcm in DN IKLK decreases. The data potentially suggest that CA IKKB caused enhanced T cells exit into the draining lymph nodes. No discussion or further experiments were performed to test this possibility.

4) In figure 5, they show that, using an in vitro system, CA IKKB reduces pSMAD2 and increases SMAD7 expression splenic T cells with Granzyme B Cre mice. Stimulation was 2 days in vitro. This is a very different condition from what they showed earlier in vivo. No data on the status of TGF beta signaling is shown. The link between TNF-NF-kB- and inhibition of TGF-beta has been already published (PMID: 10652273). Thus, the main issue here is if their contiguous stimulation of NF-kB in Trm causes TGF-beta signal inhibition in Trm cells AND if that is responsible for the reduction.

5) Their entire system relies on the use of mutant IKKB expression in T cells. While this approach shows the potential role of NF-kB in Trm development, it does not clarify how such NF-kB activation takes place in vivo. They show that TNF can induce similar results in the intracellular events in Fig. 5 in vitro. If so, do they observe a decrease of Trm by blocking TNF in vivo during the same period of infection?

Minor points

- 1) A majority of quantitation of protein/gene expression is relying on a semi-quantitative method using MFI from the flow cytometry. It is essential to confirm the results using more rigorous methods using q-PCR and western blot (with quantification along with that loading controls). Fig.5 western blot has to be done with the relative amount of each band against loading controls and performed more than once.
- 2) Fig. 2. There is a marked reduction of CD4 T cells in CA-IKK mice, though it is not reaching statistical differences. The authors describe there is no difference, but this is not true. There should be some considerations and discussion if this decrease of CD4 is affecting Trm. Statistical differences do not establish or deny biological differences.
- 3) Figure 3 has many issues that need to be clarified. (1) Why do they use VSV, not influenza, in this set of the experiment? No rationale. Since they are not using the doxy-inducible system, the way IKKB is activated is different from the flu-infection system they use. No discussion on this point. (2). A more significant point. In male mice, they see a significant reduction in Trm in CA-IKK T cells. However, they do not observe any differences in the viral titer. Why?
- 4) Overall, the writing style is too succinct and description of the results is limited. This makes it difficult for the readers to evaluate the relevance of each data presented.

RESPONSE TO THE REVIEWERS. We thank the Reviewers for the positive and constructive comments. We have made a genuine effort to answer all their critiques and believe the strength of the manuscript has been reinforced. Please see below our detailed answers to the critiques.

Reviewer #1 Trm, T cells, virus infection (Remarks to the Author):

An inducible tetOn expression system has been used to determine how temporal changes in IKK2/NFkB signaling influence survival of TRM cells in the lungs during infection with influenza A virus and VSV. Constitutively-active and dominant-negative forms of IKK are expressed in CD2+ cells. The authors report that the level of NFkB signaling inversely correlates with the numbers of pathogen-specific TRM cells in the lungs. The study further shows that deoxycycline-dependent induction of IKK2 results in upregulation of NFkB dependent genes CD69 and Eomes. Although novel technology is used to make several interesting observations, the data are not very robust. Enthusiasm of the study is reduced by an over-reliance on IV staining to identify TRM cells. No figures show TRM cells collected from the lungs by bronchoalveolar lavage. Several figures show very low frequencies of TRM cells, that may be insufficient for reliable statistical analyses. Frequent changes between infection models makes data interpretation challenging. The rationale for some experiments is unclear.

IV staining is used widely in the T cell memory field to evaluate T_{RM} as discussed in Masopust et al. Annual Rev. of Immunology, 2019; Farber and colleagues, Science Immunol. 2019; Klonowski and colleagues. Front. Immunol. 2018. Most of the articles in the field use this method to evaluate memory T cells that reside in the parenchyma (T_{RM} if they are antigen specific and found upon antigen priming) or CD69 surface staining (bona fide T_{RM}). No method is infallible as parabiosis also presents its own drawbacks (discussed in Masopust et al. Ann. Rev. Immunol. 2019). We chose this method because is widely accepted and used by researchers in the field (a search in Pub med using intravascular labeling and tissue resident memory t cells as keywords, showed 220 research articles in the last 5 years).

Our total numbers align with the T_{RM} numbers described by others in the literature. The value of the frequencies depends on the gate strategy used but as shown in the dot plot is based on detection of discrete and clear CD8 T cell populations. Please note these are polyclonal immune responses detected based on one immunodominant influenza epitope so low frequencies are expected. Indeed, for that reason, numerous studies used monoclonal TCR transgenic models (OT-1, P14) and influenza virus expressing a cognate antigen (OVA or gp33). When we used the OT-1 model in our studies, frequencies and numbers are higher but the results using both polyclonal and transgenic models are the same.

Only 2 infection models are used in this manuscript. Influenza and Vesicular stomatitis virus respiratory infections. We believe the confirmation of results using both models is more an asset than a problem. It indeed emphasizes the robustness of the conclusions as they replicate in two independent models.

1) Line 90-93. That authors indicate that NFkB signaling was manipulated “after the peak of the T cell response” and “during the contraction phase”. A relevant figure is not indicated. Does this simply mean continuous treatment between 5-30dpi? This question arises again on line 123.

Line 105 states that 5 dpi is peak of the CTL response to IAV infection. According to most published studies, the CTL response peaks ~9dpi.

A cartoon describing the experiment has been included in Extended Data Figure 2. Now, Supplementary Fig. 2e. See also figure legend.

We agree that peak of the T cell response to influenza happens between day 7-9 depending on the virus dose and precursor frequency (in the case of adoptive transfer models). However, induction of the transgene in the tetON model takes 3 days to be effective *in vivo*, in our hands. Thus, dox treatment at day 5 is equivalent to manipulation of NFkB signaling at ~day 8. Thus, the induction or inhibition of NFkB signaling starts around peak, as originally stated, and occurs mainly during the contraction phase of the immune response.

Regardless, the reference to the peak of the response has been removed to avoid confusion (original manuscript, lines 91 and 105). This is highlighted in yellow in lines 103-105 of the new version of the manuscript.

2) Some figures show very low frequencies of Tetramer+ TRM cells (less than 0.1%). Figure 1 should include flow plots show tetramer staining in the mutant mice plus/minus dox, as well as the numbers of PA-specific TRM cells in the lungs.

Frequencies shown are low because they were calculated over the single cell lymphocyte gate. This is what appears in the graphs and is shown as representative value in the dot plots in current Fig. 1. We have included the frequencies over the CD8 gate and IV gate in current Supplementary Figure 3. Our total numbers fall along with the numbers that other influenza investigators have published for influenza infection in B6 mice (note this is a polyclonal response): Zens and Farber, JCI,2016; Goplen and Sun. Science Immunol 2020. Clarification of the gate used is now included in all figure legends.

We have included dot plots for PA-specific T_{RM} s in Figure 1e and Supplementary Fig. 3d.

3) Line 115. The authors indicate that IAV specific CD8 TRM cells were found in spleen 30dpi. However, these cells are incompletely characterized. Although IV staining can distinguish cells in the red and white pulp, this technique only provides a snap-shot view of the CTL response and does not provide evidence of prolonged residence in a specific tissue. There is concern that the staining was not 100% efficient and that some circulating cells were not stained by the injected antibody. To conclude that these splenocytes are indeed TRM cells, the authors should analyze canonical markers of TRM cells (CD69, CD103, Hobit) and study migration by parabiosis.

The reviewer is correct. The complex nature of the tissue residency in the spleen was not addressed as this is beyond the scope of this paper. This was shown as an example of what happened in other tissues. The idea that NFkB signaling affects T_{RM} across different tissues was not only based on spleen. It was confirmed in kidney as shown by our results in Fig.1g. Of note, other influenza researchers also look at T_{RM} or parenchyma resident memory T_{RM} in the spleen (Jie Sun's group Sc. Immunol. 2020).

4) Since CD2 is expressed on NK cells and DCs, the CTL response may have been altered by off-target effects on other types of cells. Since inflammation impacts T cell-migration/survival, authors should verify whether prolonged Dox treatment causes immune pathology and inflammation.

This has been addressed in current Supplementary Fig.4. Histopathology concluded long exposure to dox treatment did not result in lung inflammation. Line 126-132, in yellow in current version of the manuscript. Neither NK nor DCs from dox treated animals exhibited an over activated profile indicating the results are not due to off-target effects of dox treatment on NK or DCs. Regardless, the adoptive transfer of IKK2 transgenic CD8 T cells into non transgenic hosts experiments in Fig. 2d-f clearly show that the impact of NFkB signaling on T_{RM} is CD8 T cell intrinsic.

5) Line 119. The authors state that they “did not observe important defects in the lung IAV-specific CD4 T cell memory compartment”. No functional or phenotypic data are provided. Figure 2 does not clearly indicate how many mice were analyzed (six?), however an apparent reduction in the numbers of CD4 T cells (Fig 2C) may be significant if larger cohorts are used.

We agree with the reviewer that this is a very general statement that can lead to inaccurate interpretations. To solve this, we have clarified our statement in Lines 122-125 of the current version of the manuscript: “We also did not observe differences in the frequency of IAV-specific CD4 memory T cells when NFkB signals were increased, suggesting that the defect in CD8 T_{RM} was not due to decreased or increased generation of flu-specific CD4 T cells (Supplementary Fig.4d).”

A potential role of NFkB signaling in the differentiation and function of antigen specific CD4 T cells, while interesting, is beyond the scope of these studies. Again, adoptive transfer experiments in Fig 2, clearly show that the impact of NFkB signaling in CD8 T_{RM} is CD8 T cell intrinsic.

6) Line 122. The authors indicate that they used an adoptive transfer model to confirm whether the effect of NFkB signaling was CD8 T cell intrinsic. Figure 3 shows that increased NFkB signaling augments T cell effector function (Fig 3e). Since proinflammatory cytokines can indirectly induce T cell death by activating myeloid cells in the local tissues, the loss of TRM cells may be due to extrinsic inflammation.

Please note that the data in Figure 3 refers to secretion of cytokines after recall of memory T cells *ex vivo* (once they have been formed) and show that when CA-IKK2^{ON} T_{RM}s are re-stimulated by antigen, the frequency of lung memory T cells able to express TNF and IFN γ is increased compared to controls. This can be explained because NFkB signaling is important for memory effector function as described in Lai et al. The Journal of Immunology. 2011.

Furthermore, we reasoned that overt inflammation provided by inflammatory cytokines, even if secreted by T_{RM}, would cause unspecific T cell death, which is clearly not the case for central or memory T cells (Figure 1,6) or non-antigen specific T cells in the lung (Figure 2b,c)

7) Line 133. The authors state that a published approach was used to deplete circulating CD8 T cells, while sparing CD8 TRM. No reference is provided. As male mice are generally larger than females, there is concern that gender might influence disease progression.

This is a well-established method used widely in the field to eliminate circulatory T cells based on controlled rejection. The reference was provided (no. 45, now no. 46, line 148) in the original article, Gebhardt et al. Nature 2011.

8) Figure 4. It is unclear why CD69 and CD103 were analyzed separately. Most mucosal TRM cells express both markers, as indicated on line 50.

We wanted to analyze both markers independently as studies have reported that in the lung there are CD69+CD103+ CD69+CD103- populations (Kohlmeier's group Nature Immunology 2020; Wu et al. Cell reports 2020)

9) The authors should discuss the relevance of Nur77 expression. Does this marker indicate a response to persisting antigen?

We are aware that Nur77 is used in the literature as a downstream marker of TCR signaling. However, its role extends beyond this. Nur77 can be induced independent of antigen stimulation and has been linked to cell survival (Suzuki et al. PNAS 2003). Most importantly, Nur77 has been linked to T_{RM} differentiation (Boddupalli et al. JCI 2016 and Allan and colleagues Cell Reports 20210). Since we did not manipulate antigen dose, affinity or TCR signal duration in our experiments, we do not consider it pertinent to discuss its role in the TCR/ persistent antigen scenario. Rather, we favor the idea that Nur77 is a marker of T_{RM}, whose expression is altered by NFκB signals. Interactions between Nur77 and NFκB have been described at length in the literature, mostly in non-T cells. How this cross-talk leads to the regulation of T cell memory is interesting and, undoubtedly, requires further investigation beyond the scope of the studies here.

10) The manuscript includes some typographical errors. **We hope these have been addressed in the current version.**

Reviewer #2 T cells (Remarks to the Author):

The authors examine the impact of enhancing and inhibiting NFκB signaling on generation and maintenance of distinct memory CD8+ T cell subsets. Using constitutively active (CA) and dominant negative (DN) IKK2 constructs under a tetracycline inducible genetic system (or, in some experiments, Cre-mediated induction), the authors find a series of interesting findings. First, CA IKK2 induction following the peak expansion of the CD8+ T cell response promotes generation of recirculating "central" memory CD8+ T cells yet inhibits generation of "resident" memory CD8+

T cells (TRM) in the lungs (influenza infection) and kidney (VSV infection). This includes dysregulated expression of several genes associated with TRM, including reduced CD103, Runx3, Nur77 and CD122, but increased Eomes expression. Expression of DN IKK leads to reciprocal changes for the most part. The authors show that signals through TGF- β (which induce CD103 and are key to establishing TRM) are antagonized by NF κ B stimulating factors such as TNF α . Interestingly, the authors find that the timing of NF κ B regulation critically affects outcomes – induction of CA IKK2 after CD8⁺ T cell memory has already formed leads to improved (not reduced) persistence of TRM, indicating that the roles of NF κ B in control of TRM change during generation and maintenance.

The authors have developed elegant and powerful mouse models to dissect the impact of NF κ B signaling on the fate and homeostasis of CD8⁺ T cell memory populations, building on their previous studies showing a role for NF κ B signals in memory T cell preservation. While the findings are complicated (reflecting the underlying biology), the message is clear, the data compelling and the studies important for understanding the role of the NF κ B pathway in shaping T cell memory. In particular, the authors findings suggest a potential role for diminished NF κ B signaling as a cause of the accelerated decay in the lung TRM population (compared to TRM in other tissues), which is well known but poorly understood.

There are some concerns that will need to be addressed, however.

- 1) The authors focus on the IKK2 CA model, with less data on the IKK2 DN transgenic mice. While there is no need to repeat all studies in both models, it would have been powerful for the authors to show the impact of cell-intrinsic effects of IKK2 DN induction in generation of TRM (essentially following the approach outlined in Fig. 2e). This would allow the authors to determine whether effects of both increased and decreased NF κ B are CD8⁺ T cell-intrinsic.

We appreciate the insightful comment of the reviewer. We now provide results regarding the cell intrinsic effect of IKK2DN in the generation of T_{RM} (current Fig. 2f, Lines 136-138 in yellow) and T_{RM} maintenance (current Figure 6f-j, Lines 299-303 in yellow).

“... inhibiting NF κ B signaling intrinsically in CD8 T cells, led to an increase in the generation of influenza-specific lung CD8 T_{RM} (Fig. 2f)”.

- 2) Along the same lines, it would be interesting to see whether induction of IKK2 DN in the established memory CD8⁺ T cells would lead to accelerated loss of TRM (i.e. use of the DN model as in Fig. 6). This reviewer appreciates, that such studies would take considerable time, however, but some consideration of this issue would be valuable.

We have included data from the experiments the reviewer suggested in Fig. 6f-j. Lines 298-302 in yellow in the main text: “We also performed similar experiments using the DN-IKK2^{DN} inducible model. Inhibiting NF κ B signaling once have formed, had the converse effect (Fig. 6f-j) a loss in CD8 T_{RM} numbers and a decrease in the expression of survival factors (CD122 and Bcl2). This was similar for the circulatory T cells in lung and for T_{CM} cells in draining lymph nodes, further demonstrating that NF κ B signals are key for the survival and maintenance of CD8 memory upon influenza infection”.

- 3) Do the authors have any data on whether enhanced maintenance of lung TRM by late IKK2

CA induction may reflect increased proliferation (in addition to the proposed Bcl-2 related avoidance of cell death)? BrdU incorporation assays could be used to assess this.

These data are included in Supplementary Fig. 7. Lines 274-283, highlighted in yellow in the main manuscript. We performed BrdU incorporation experiments at memory and found no differences between control and mice where NFkB signaling was enforced. However, in the same experiments, CA-IKK2ON cells exhibited higher levels of pro-survival factors CD122 and Bcl2 than controls. From these, we conclude that NFkB signaling does not support T_{RM} survival through increased turnover or proliferation. Curiously CA-IKK2ON circulatory T cells in the lung also exhibited higher levels of Bcl2, suggesting at memory the main role of NFkB is to support survival across all memory T cell subsets (discussed in lines 274-283)

4) The authors carefully describe the loss of parenchymal (iv antibody -ve) cells in the lung, and also show reduced numbers of TRM- phenotype cells when IKK2 CA is induced. However, it is less clear whether there is a change in the numbers of recirculating cells in the lung parenchyma by IKK2 CA (or DN) expression – i.e. whether there is an increase or decrease in the tissue-trafficking “effector” memory population. This could be shown as the frequency and numbers of CD69-ve antigen-specific cells in the iv antibody -ve pool (perhaps these data could be added to Fig. 4c).

We thank the reviewer for raising this comment as we also find it intriguing. The analysis the reviewer requested are now included in Figure 4c and h. and Lines 173-178, 189-192 in the main text highlighted in yellow: “When we assessed re-circulating cells within the lung parenchyma by frequency and number of CD69 negative within the iv negative CD8 population, we observed that CA-IKK2^{ON} CD8 T cells lost CD69 expression and remained in the parenchyma, likely increasing the trafficking “effector” memory CD8 population (Fig. 4c, day 30, right graphs).

We conclude that enhanced NFkB signals do not interfere with the recruitment of CA-IKK2^{ON} cells to the lung but rather with the ability of these cells to stay in parenchyma as “bona fide” T_{RM} (CD69⁺): “Importantly, CD69 negative CA-IKK2^{ON} CD8 T cells that remain in the parenchyma also exhibited defects in the expression of T_{RM} transcription factors Eomes and Runx 3 when compared with their CD69 positive counterparts and WT controls (Fig. 4h)”. This suggests to us that CA-IKK2^{ON} cells are impaired in achieving the transcriptional profile necessary to commit to T_{RM}.

5) Have the authors any data on whether transient induction of IKK2 CA during memory maintenance (as in Fig. 6) leads to sustained increase in the lung TRM pool? Or, instead, does that population “crash” once IKK2 CA expression is allowed to decline? This relevant as to whether the provision of increased NFkB signaling “rescues” lung TRM maintenance or only provides a temporary reprieve in survival.

We have performed experiments to assess this comment. The data is shown in Supplementary Figure 8 and discussed in lines 287-298 in the text, highlighted in yellow: “...we tested whether transient induction of IKK2/NFkB signaling at memory would provide a temporary or a long-term increase in IAV specific -CD8 T_{RM}. To test this in a CD8 T cell intrinsic manner, we transferred OT-1 naïve donors from OT-1CD90.1xCD2rtTAXCA-

IKK2 mice into CD90.2 B6 host and infected the mice with PR8-OVA. At memory, we treated chimeric mice with or without doxycycline for 10 days as in Fig 6a and then removed treatment in a cohort of the dox treated mice and not in the other. 15 days after interrupting the induction of NFkB signaling (dox treatment), we assessed IAV-specific CD8 T_{RM} decay. While continuous provision of NFkB signal led to a steady increase of lung CD8 T_{RM}, removal of IKK2 constitutive signaling reverted the numbers of CD8 T_{RM} to control levels (Fig S9). From these, we concluded that an increase in NFkB signaling at memory can boost lung CD8T_{RM}, as long as, the provision of NFkB signaling is continuously maintained. Interruption of NFkB signaling also affected lung circulating memory, most likely because NFkB signaling supports survival of all CD8 memory subsets”.

Reviewer #3 NFkb, IKK (Remarks to the Author): Summary of the paper

This manuscript describes new observations of lung residential CD8 memory T cells in transgenic mice that express a constitutively active form and a dominant-negative form of Ikb kinase B (IKKB). The authors infect these mice with influenza A virus or VSV expressing OVA and measure the level of antigen reactive CD8 T cells in the lung tissue using tetramers. When they induce a constitutively active form of IKKB from 5 days after infection till day 30, they observe a significant reduction in tetramer-positive lung residential CD8 T cells. Conversely, they observe an increase in tetramer+ CD8 Trm cells in the lung when a dominant negative form is expressed. To understand the mechanism of these observations, they further analyze the cells that remain in the CA-IKKB transgenic mouse and found that the frequency and cell numbers of the tetramer+ cells expressing CD69 and CD103 are reduced in these mice. Using an in vitro system, the authors show that TGF-beta signaling is inhibited by the active form of IKK B and TNF, suggesting that TNF may be a potential factor that can activate NF-kB during inflammation, blocking TGFbeta signaling, and causing a reduction in CD8 lung residential memory T cells.

Comments

The most noteworthy result from this work is shown in Figure1 and Figure 6 whereby expression of the active form of IKKB have opposite effect on CD8 lung Trms depending on the timing of expression. Contrary to the day 5-30 induction of CA IKKB, post day30 expression of CA IKKB increases the number of tetramer-positive T cells.

While these observations are novel and potentially significant for the field, the current manuscript does not provide sufficient results to support their claim. Followings are the major issues that need to be addressed.

- 1) In Figure 6, the authors show that activation of NF-kB 30 days after infection enhances the CD8 TRM. They claim this suggests that the Nf-kB signal supports the expansion of Trm in the late phase of infection. This is a puzzling observation because the experiment is designed such that no antigen stimulation is presumably present. If so, why only tetramer-positive Trm

cells expand? There are other CD8^{Trm} exist in the lung. The data show over 99% of Trm cells is tetramer negative. The induction of CA IKKB is driven by CD2, thus all T cells are expressing active IKKB. Why the other Trm cells do not expand in response to the induction of CA IKK? This point requires rigorous clarification.

We agree that no antigen stimulation is present in these assays as the experiments only involve the treatment with doxycycline and no re-infection. We never claimed that the memory cells expand but rather our data suggested that they survive better (see pro-survival data provided using CD122 and Bcl2 as readouts of factors that support T cell survival). Lines, 230-232 of original manuscript state: “The increase in CA-IKK2^{ON} CD8 T_{RM} at memory correlated with higher levels of CD122 and Bcl-2, suggesting NFkB signals at memory mediate CD8 T_{RM} survival (Fig 6b).

Nevertheless, we have performed experiments addressing a potential impact of NFkB signaling in expansion, turnover or homeostasis of CD8 T_{RM}. These data are included in Supplementary Fig. 7. Lines 274-283, highlighted in yellow in the main manuscript: “We performed BrdU incorporation experiments at memory and found no differences between control and mice where NFkB signaling was enforced. However, in the same experiments, CA-IKK2^{ON} cells exhibited higher levels of pro-survival factors CD122 and Bcl2 than controls. From these, we conclude that NFkB signaling does not support T_{RM} maintenance through increased turnover or proliferation”.

- 2) In supplementary Figure 2, they show that expression of CA IKKB does not induce caspase 3 activation or FasL expression (surface stain), and conclude that this transgene expression does not induce cell death. However, there is no *in vivo* data that support this assumption. This manuscript needs to determine if constitutive and sustained IKKB active form expression is causing accelerated cell death, migration (discussed below), or blocking differentiation as they claim. It has been shown at TGF-beta not only induces differentiation but also inhibits cell death by many systems.

The experiments in supplementary Fig. 2 were performed *in vivo* as indicated in the Figure legend:” (d) Expression of markers indicated in CD8 T cells of CD2rtTA x CAIKK2 mice (green) or negative littermates (grey) that have been treated with dox solution for 7 days.” These experiments were designed to determine whether constitutive induction of NFkB signaling triggered by the administration of dox to mice, could cause generalized, overt, non-antigen-specific T cell death. This is clearly not the case as shown multiple times in this study, including Supplementary Fig. 2d and Figure 2b, c. Collectively, these data indicate that enhanced NFkB signaling affects only antigen specific T cells.

In addition, the fact that cell loss does not happen for T_{CM} or T_{EM} (Figure 1) negates that enhanced NFkB signaling is inducing a general loss of all T cell subsets. Instead, it specifically affects antigen specific T_{RM}.

Most importantly, experiments in Figure 4 clearly demonstrate that NFkB signaling impacts the differentiation of T cells towards the T_{RM} fate as the T_{RM} transcriptional program is impaired. Indeed, the expression of T_{RM}-associated transcription factors Runx3, Eomes and Nur77 is altered in CA-IKK2^{ON} cells at day 30 (Figure 3g, h). We also conclude that in addition, NFkB signaling may be affecting T_{RM} survival as IL-15R (CD122) expression is also impaired.

The fact that we preferentially loss CD69+ bona fide T_{RM} and that there is no loss in lung flu-specific circulatory T cells (or defects in CXCR3) argues against the idea of a defect in migration to the lung.

In conclusion, we agree with the reviewer that our data shows NFkB signaling is counteracting TGFb'role in T_{RM} differentiation and T_{RM} survival, which is a novel finding.

3) In Fig.1b, the data show that Tcm in the mediastinal lymph nodes in CA KKK mouse significantly increases while Tcm in DN IKLK decreases. The data potentially suggest that CA IKKB caused enhanced T cells exit into the draining lymph nodes. No discussion or further experiments were performed to test this possibility.

We appreciate the insight of the reviewer and considered the suggestion interesting. However, we believe this is an unlikely possibility. We reasoned that if the increase in T_{CM} in LN would be caused by an increase in recruitment of T cells back into LN, we would have seen changes in the numbers of circulating T cells and/or a parallel increase in T_{EM} in the dLN, which is not the case. Instead, the fact that NFkB signaling regulates the expression of the T_{CM} master regulator Eomes better justifies the increase in T_{CM}, as our previous published studies have shown in Knudson et al. PNAS2017.

4) In figure 5, they show that, using an in vitro system, CA IKKB reduces pSMAD2 and increases SMAD7 expression splenic T cells with Granzyme B Cre mice. Stimulation was 2 days in vitro. This is a very different condition from what they showed earlier in vivo. No data on the status of TGF beta signaling is shown. The link between TNF-NF-kB- and inhibition of TGF-beta has been already published (PMID: 10652273). Thus, the main issue here is if their contiguous stimulation of NF-kB in Trm causes TGF-beta signal inhibition in Trm cells AND if that is responsible for the reduction.

We did cite this reference in our original manuscript, but we prompt the reviewer to notice this was reported in fibroblasts and, not necessarily, would be expected to operate in the same manner in all cell types. To our knowledge, no study has shown this mechanism operates in T cells and most importantly regulates CD8 T_{RM} in the context of infection.

To rigorously assess the crosstalk between NFkB and TGFb signaling, the experiment variables need to be controlled. We reasoned that to mimic the differentiation status of the T cell *in vivo* and the unique interaction between TGF and NFkB signaling two variables needed to be under control. (1) Activated T cells differentiating to effectors (48 h in vitro stimulation) and (2) singular exposure to TGFb in the presence or absence of enhanced NFkB signaling. Thus, the data in Fig. 5 a), b) and d) unequivocally demonstrate that NFkB signaling regulates TGFb signaling. The status of TGFb signaling is indeed shown and measured as the induction of phosphorylated-Smad2/3 by flow and western blot. Assessing active TGFb receptor signaling upon TGFb stimulation by measuring the phosphorylation of Smad2/3, It is well accepted in the field. The status of TGFb signaling is shown for resting and unstimulated cells as controls.

We have added data of the crosstalk of NFkB and TGFb signaling *in vivo*. This is now shown in Figure 7c. In the figure, it is shown that at day 7p.i. CA-IKK2ON cells in the lung are impaired in the phosphorylation of Smad2/3 (aka TGFb signaling) while exhibiting higher levels of NFkB signaling (phosphorylated-p65) than control.

5) Their entire system relies on the use of mutant IKKB expression in T cells. While this approach shows the potential role of NF- κ B in T_{RM} development, it does not clarify how such NF- κ B activation takes place *in vivo*. They show that TNF can induce similar results in the intracellular events in Fig. 5 *in vitro*. If so, do they observe a decrease of T_{RM} by blocking TNF *in vivo* during the same period of infection?

We thank the reviewer for this insightful suggestion. Experiments to assess the concern of the reviewer were performed *in vivo* and are now shown in Figure 5j-m. Lines 239-254 in yellow in the main text.

The *in vitro* results in Fig. 5e-h, suggest that TNF-dependent NF κ B signaling inhibits TGF β -dependent T_{RM} differentiation. Since TNF levels are induced upon influenza infection and TGF β in lung tissue regulates T_{RM}, the expectation would be that high levels of TNF would lead to a decrease in T_{RM} (and T_{RM} associated transcription factors) and that a blockade in TNF, would lead to a recovery or an increase in T_{RM}. This is what we found, and it is now shown in Figure 5j-m. Interestingly, blockade of TNF *in vivo*, led to a recovery of pSMAD2/3 and Runx3 levels, confirming the *in vitro* results.

Minor points

1) A majority of quantitation of protein/gene expression is relying on a semi-quantitative method using MFI from the flow cytometry. It is essential to confirm the results using more rigorous methods using q-PCR and western blot (with quantification along with that loading controls). Fig.5 western blot has to be done with the relative amount of each band against loading controls and performed more than once.

Densitometry was originally provided in Supplemental Figure 6 (current Supplementary Fig.6). In the figure legend, appears clearly stated that the experiments were representative of several independent experiments. Not only that, the results in Fig 5d, were repeated in two independent inducible CA-IKK2ON models (See Supplementary Figure 6).

2) Fig. 2. There is a marked reduction of CD4 T cells in CA-IKK mice, though it is not reaching statistical differences. The authors describe there is no difference, but this is not true. There should be some considerations and discussion if this decrease of CD4 is affecting T_{RM}. Statistical differences do not establish or deny biological differences.

Since there is no statistical difference, we cannot conclude that there are differences in the frequency of flu specific CD4 T cells that are linked to the defect in T_{RM}. Nevertheless, we have clarified our statement in Lines 122-125 of the current version of the manuscript: “We also did not observe differences in the frequency of IAV-specific CD4 memory T cells when NF κ B signals were increased, suggesting that the defect in CD8 T_{RM} was not due to decreased or increased generation of flu-specific CD4 T cells (Supplementary Fig.4d).”

A potential role of NF κ B signaling in the differentiation and function of antigen specific CD4 T cells while interesting is beyond the scope of these studies. Again, adoptive transfer experiments in Fig 2, clearly show that the impact of NF κ B signaling in CD8 T_{RM} is CD8 T cell intrinsic.

3) Figure 3 has many issues that need to be clarified. (1) Why do they use VSV, not influenza, in this set of the experiment? No rationale. Since they are not using the doxy-inducible system, the way IKKB is activated is different from the flu-infection system they use. No discussion on

this point. (2). A more significant point. In male mice, they see a significant reduction in Trm in CA-IKK T cells. However, they do not observe any differences in the viral titer. Why?

- **VSV-OVA intranasal infection and PR8-OVA recall is used as a model of heterologous immunity to rigorously assess the OVA specific TRM function in an antigen specific manner. This way, the impact of pre-existing immunity by endogenous non-OT-1 cells can be discarded and the results can truly be accounted by antigen-specific CD8T_{RM}.**
 - **The OT-1xIKK2CA^{fl/fl}xGzB^{Cre} was chosen because it allows us to specifically assess the effect in a monoclonal TCR tg model. Indeed, we consider that obtaining the same results (Fig3d) in two models adds rigor and value to our studies as same results are recapitulated in two independent inducible models (CA-IKK2ON and the IKK2CAfl/flxGzBcre models).**
 - **The T_{RM} response to flu is measured in this system as titers of influenza virus remaining after PR8-OVA infection (Fig.3e). Another way to understand these experiments is in the context of a response to influenza infection upon immunization with an unrelated virus (similar to vaccination).**
 - **We interpret the lack of difference in viral titers in male mice as the contribution of both circulatory and T_{RM} cells while in the female mice only lung T_{RM} can contribute - as the circulatory T cells have been depleted-. Please note that CA-IKK2ON cells are not defective in antigen specific T_{CM} or T_{EM} or lung circulatory T cells (Fig. 1 and 4)**
- 4) Overall, the writing style is too succinct and description of the results is limited. This makes it difficult for the readers to evaluate the relevance of each data presented.

Unfortunately, the word limitation prevents us from further explanation. The models and methods used are widely employed, known and accepted in the field.

REVIEWER COMMENTS

Reviewer #1 (Remarks to the Author):

Although this revised manuscript is considerably improved, some of my prior concerns have not been fully addressed and I am not persuaded by discussion of other literature. Although IV staining is widely used to study TRM cells, this method does not indicate how long cells remain in specific tissues. When IAV infected mice were analyzed by parabiosis, donor and recipient CTLs rapidly equilibrated in the spleen, but not the lungs or MLN. These studies indicate that the spleen does contain many TRM cells after IAV infection. IV staining cannot distinguish resident cells from circulating cells that are transiting through the spleen. Similarly, IV staining is not suitable for analyzing migrating cells in the MLN (Fig 3B) since circulating T cells (naïve and TCM) enter the cortex and become inaccessible to injected antibodies. Clarification is required at multiple places in the paper. Predominant populations of pulmonary TRM cells express both CD69 and CD103 after IAV infection, consequently the IV staining would be more convincing if some flow plots in the lungs showed CTLs expressing both markers.

Figure 1. The legend indicates that TEM cells were identified using CD44 and CD62L expression (lines 643-644). Since TEM lack lymphoid homing receptors, they are mostly excluded from SLO. I suspect that the MLNs contain substantial numbers of lymphoid TRM cells that expressed CD103/CD69, as reported previously.

Figure 1. Presentation of the bar graphs remains a concern. Graphs showing percentages of cells within a live gate are uninformative, since individual mice show considerable variation in other populations of immune cells. In the same figure, contour plots show gated CD8+ T cells but the percentages of tetramer+ cells are still surprisingly low. The discrepancy requires clarification. What were the frequencies of Tetramer+ cells in Figure 1G?

Some bar graphs could be labeled more clearly - Figure 2D-F.

Supplementary Figure 2B. No error bars are shown. How many mice were analyzed?

Supplementary Figure 6. Since three experiments were performed, the bar graphs should show combined data with error bars.

Does constitutively active IKK2 impact thymic development or survival of naïve CD8 T cells?

Lines 173-178. This section is confusing. If CD8 T cells lose CD69 expression and remain in the parenchyma, why do the authors suggest that they are trafficking? What is the mechanism of tissue retention? Do they express CD103? Some clarification is required

Some grammatical errors need correcting.

This report was prepared by Linda Cauley Ph.D.

Reviewer #2 (Remarks to the Author):

The authors have done an excellent job of revising the manuscript in response to previous concerns -- this includes new data that are compelling and support key elements of the authors' interpretation. No remaining concerns.

Reviewer #3 (Remarks to the Author):

All the concerns were addressed properly in this revision. I recommend accepting the manuscript for publication.

RESPONSE TO REVIEWER NO. 1

Although this revised manuscript is considerably improved, some of my prior concerns have not been fully addressed and I am not persuaded by discussion of other literature. Although IV staining is widely used to study TRM cells, this method does not indicate how long cells remain in specific tissues. When IAV infected mice were analyzed by parabiosis, donor and recipient CTLs rapidly equilibrated in the spleen, but not the lungs or MLN. These studies indicate that the spleen does contain many TRM cells after IAV infection. IV staining cannot distinguish resident cells from circulating cells that are transitioning through the spleen. Similarly, IV staining is not suitable for analyzing migrating cells in the MLN (Fig 3B) since circulating T cells (naïve and TCM) enter the cortex and become inaccessible to injected antibodies. Clarification is required at multiple places in the paper.

We agree with the reviewer, IV staining may not be an accurate methodology to identify spleen TRM cells. For that reason, we have decided to remove this data from the manuscript. Furthermore, we have stated the limitations of the IV staining in the discussion, lines 368-376.

The data presented for TCM and TEM in Fig. 1 in the last version of the manuscript did not distinguish between IV positive or IV negative cells but rather represented CD44hi CD62Lhi or CD44hi CD62Llo co-expressing cells of total influenza specific CD8 T cells in the whole MLN. This is the way TCM and TEM are defined and refer to in the field. Our intention was not to define TRMs in the MLN.

As to the data in Fig 3b and c. There was an error in the figure legend that has now been corrected. We apologize for the confusion. The data in Fig 3b was not gated in IV+ cells but in total cells in the MLN. We have replaced the graph in 3c for data in the spleen (no IV labeling gate) to better reflect efficacy of circulatory T cell rejection at day 30 p.i. Importantly, the new data shows the same differences as before.

- Predominant populations of pulmonary TRM cells express both CD69 and CD103 after IAV infection, consequently the IV staining would be more convincing if some flow plots in the lungs showed CTLs expressing both markers.*

Gated on IV⁺ CD8⁺ D⁺-NP₃₅₆ Tet⁺ cells

We thank the reviewer for this comment. We now provide this data in new Fig. 1f and Figure 4d, Supplementary Fig. 6. Increasing IKK2 signaling leads to a significant decrease in the frequency and number of T cells co-expressing CD69 and CD103 in the lung parenchyma. Conversely, decreasing IKK2 signaling increases the frequency and number of T cells co-expressing CD69 and CD103 in the lung parenchyma. These differences match the differences that we reported with the IV labelling strategy.

Additionally, we would like the reviewer to consider the data in Fig. 3d as well, looking at TRM with another method used in the field (van de Wall and Harty; cshperspect.a037978): After depletion of circulating T cells (in female hosts) the frequency(3d) and number (Supplemental Fig5b) of CA-IKK2ON cells in the lung parenchyma/TRM is significantly lower than controls. Furthermore, if the decrease in CA-IKK2ON memory cells in the lung parenchyma was only due to TEM cells circulating in tissue, we should have seen defects in EM recruitment or generation in SLO and we do not.

- *Figure 1. The legend indicates that TEM cells were identified using CD44 and CD62L expression (lines 643- 644). Since TEM lack lymphoid homing receptors, they are mostly excluded from SLO. I suspect that the MLNs contain substantial numbers of lymphoid TRM cells that expressed CD103/CD69, as reported previously.*

We do not have any comment regarding this. We are following the nomenclature that has been established in the field for a long time. Quantifying tissue resident memory T cells in the MLN (including retrograde migration) is outside of the scope of these studies.

- *Figure 1. Presentation of the bar graphs remains a concern. Graphs showing percentages of cells within a live gate are uninformative, since individual mice show considerable variation in other populations of immune cells. In the same figure, contour plots show gated CD8+ T cells but the percentages of tetramer+ cells are still surprisingly low. The discrepancy requires clarification. What were the frequencies of Tetramer+ cells in Figure 1G?*

We thank the reviewer for this comment. While mathematically our data was correct, as we calculated the % over total live cells and compared it to the internal control, we understand and appreciate the concern of the reviewer in the light of how other authors in the field present data as % of total CD8 T cells or % of IAV-NP specific CD8 T cells (Wein et al. JEM 2019; Hayward et al. 2020). For that reason, we have carefully revised our bar graphs and now present data in figures and supplemental figures as % of total CD8s or % of influenza specific CD8 T cells. Our frequencies are similar to the ones described in the field following a similar approach (i.e. Wein and Kholmeier JEM 2019; Hayward et al. Nat. Immunol 2020). Most importantly, the conclusions that can be drawn from the new data are the same as what we reported in the previous version of the manuscript.

- *Some bar graphs could be labeled more clearly - Figure 2D-F. Supplementary Figure 2B. No error bars are shown. How many mice were analyzed?*

We have revised and corrected most of the axis of the bar graphs. We hope this has provided more clarity. Error bars are shown for new Fig. 2B.

- *Supplementary Figure 6. Since three experiments were performed, the bar graphs should show combined data with error bars.*

This has been addressed in new Supplementary Fig. 6 (now Supplementary Fig. 7).

- *Does constitutively active IKK2 impact thymic development or survival of naive CD8 T cells?*

As stated in the main manuscript and in Fig. 2, the experimental strategy in our studies affects only peripheral antigen specific T cells that have been activated or responded to infection for 5 days. Survival of any other CD8 or CD4 T cells that are not antigen specific is not altered as shown in Fig. 2b. The effects seen by the temporal induction of the IKK2 transgenes cannot be the consequence of alterations in T cell development in the thymus or naive T cell survival. The T cells that respond to influenza in our studies have already passed thymic selection and their development and survival as naive T cells is normal. Please consider that CD2rtTAxCA-IKK2 and CD2rtTAxDN-IKK2 mice are not subjected to DOX before infection. The role of active IKK2 in thymic development or survival is beyond the scope of this paper.

- *Lines 173-178. This section is confusing. If CD8 T cells lose CD69 expression and remain in the parenchyma, why do the authors suggest that they are trafficking? What is the mechanism of tissue retention? Do they express CD103? Some clarification is required.*

Clarification is provided in the new version of the manuscript in lines 180-185 (highlighted). These data responded to a specific request of rev. 2.

- *Some grammatical errors need correcting.*

We hope the grammatical errors have been corrected in this new version. The manuscript has been revised for grammar by the English department in the University of Missouri as well as by a British colleague in the department.

RESPONSE TO REVIEWERS 2 AND 3.

We thank reviewers 2 and 3 for their valuable time and for their recommendation for publication.

REVIEWERS' COMMENTS

Reviewer #1 (Remarks to the Author):

1) The authors state that “Our intention was not to define TRMs in the MLN”.

The following concern remains:

“ Figure 1. The legend indicates that TEM cells were identified using CD44 and CD62L expression (lines 643- 644). Since TEM lack lymphoid homing receptors, they are mostly excluded from SLO. I suspect that the MLNs contain substantial numbers of lymphoid TRM cells that expressed CD103/CD69, as reported previously”.

We do not have any comment regarding this. We are following the nomenclature that has been established in the field for a long time. Quantifying tissue resident memory T cells in the MLN (including retrograde migration) is outside of the scope of these studies.

Although I am not suggesting that additional data are required, I find the authors resistance to acknowledging that the draining lymph nodes contain TRM cells rather strange. Ignoring these cells could undermine the data interpretation and cause some confusion for readers.

2). (lines 367-375) The addition text at the end of the manuscript is out of place – should be included with the experimental IV data, or methods.

RESPONSE TO THE REVIEWERS.

Reviewers' comments

Reviewer #1 (Remarks to the Author):

1) The authors state that "Our intention was not to define TRMs in the MLN".

The following concern remains:

" Figure 1. The legend indicates that TEM cells were identified using CD44 and CD62L expression (lines 643- 644). Since TEM lack lymphoid homing receptors, they are mostly excluded from SLO. I suspect that the MLNs contain substantial numbers of lymphoid TRM cells that expressed CD103/CD69, as reported previously".

We do not have any comment regarding this. We are following the nomenclature that has been established in the field for a long time. Quantifying tissue resident memory T cells in the MLN (including retrograde migration) is outside of the scope of these studies.

Although I am not suggesting that additional data are required, I find the authors resistance to acknowledging that the draining lymph nodes contain TRM cells rather strange. Ignoring these cells could undermine the data interpretation and cause some confusion for readers.

We have made changes in "IKK2/NFkB signaling differentially regulates T cell memory subset diversity", "At memory, NFkB signaling promotes lung CD8 T_{RM} survival", discussion and figure legend sections. We have referred to CD44^{hi} CD62L^{lo} T_{MEM} populations in the MLN as such (rather than using the T_{EM} nomenclature) and mentioned that these cells may include a substantial number of T_{RM}, referencing Dr. Cauley's work.

2). (lines 367-375) The addition text at the end of the manuscript is out of place – should be included with the experimental IV data, or methods.

We have made this change and move the paragraph to the methods section ("In vivo antibody labeling and flow cytometry. ")